# Microenvironmental IL1β promotes breast cancer metastatic colonisation in the bone via activation of Wnt signalling

Rachel Eyre[1], Denis G. Alférez[1], Angélica Santiago-Gómez[1], Kath Spence[1], James C. McConnell[2], Claire Hart[3], Bruno M. Simões[1], Diane Lefley[4], Claudia Tulotta[4], Joanna Storer[1], Austin Gurney[5], Noel Clarke[6], Mick Brown[3], Sacha J. Howell[1,7], Andrew H. Sims[8], Gillian Farnie[9], Penelope D. Ottewell[4]* & Robert B. Clarke[1]*

Dissemination of tumour cells to the bone marrow is an early event in breast cancer, however cells may lie dormant for many years before bone metastases develop. Treatment for bone metastases is not curative, therefore new adjuvant therapies which prevent the colonisation of disseminated cells into metastatic lesions are required. There is evidence that cancer stem cells (CSCs) within breast tumours are capable of metastasis, but the mechanism by which these colonise bone is unknown. Here, we establish that bone marrow-derived IL1β stimulates breast cancer cell colonisation in the bone by inducing intracellular NFkB and CREB signalling in breast cancer cells, leading to autocrine Wnt signalling and CSC colony formation. Importantly, we show that inhibition of this pathway prevents both CSC colony formation in the bone environment, and bone metastasis. These findings establish that targeting IL1β-NFKB/CREB-Wnt signalling should be considered for adjuvant therapy to prevent breast cancer bone metastasis.

[1] Breast Biology Group, Manchester Breast Centre, Division of Cancer Sciences, Faculty of Biology, Medicine and Health, University of Manchester, Wilmslow Road, Manchester M20 4GJ, UK. [2] Division of Cell Matrix Biology and Regenerative Medicine, Faculty of Biology, Medicine and Health, University of Manchester, Oxford Road, Manchester M13 9PT, UK. [3] Genito Urinary Cancer Research Group, Division of Cancer Sciences, Faculty of Biology, Medicine and Health, University of Manchester, Wilmslow Road, Manchester M20 4GJ, UK. [4] Academic Unit of Clinical Oncology, Department of Oncology and Metabolism, Mellanby Centre for Bone Research, University of Sheffield, Sheffield S10 2RX, UK. [5] OncoMed Pharmaceuticals, Redwood City, CA 94063, USA. [6] Department of Urology, Salford Royal Hospital NHS Foundation Trust, Stott Lane, Salford M6 8HD, UK. [7] The Christie NHS Foundation Trust, Wilmslow Road, Manchester M20 4BX, UK. [8] Applied Bioinformatics of Cancer Group, Cancer Research UK Edinburgh Centre, Institute of Genetics and Molecular Medicine, University of Edinburgh, Edinburgh EH4 2XR, UK. [9] Structural Genomics Consortium, NDORMS, Botnar Research Centre, Windmill Road, Oxford OX3 7LD, UK. *email: p.d.ottewell@sheffield.ac.uk; Robert.clarke@manchester.ac.uk

Approximately half a million women die from breast cancer globally each year[1]. These deaths are predominantly due to metastatic disease, therefore developing new strategies to prevent or treat metastasis are essential to reduce breast cancer mortality. It is known that systemic spread is an early event in breast cancer[2,3], with disseminated tumour cells (DTCs) detectable in bone marrow at the time of primary tumour diagnosis[4]. Based on this, targeting the early steps of metastasis is unlikely to be an effective clinical strategy. However, in many cases DTCs lie dormant in the bone marrow for long periods prior to overt colonisation[5,6], and preventing this colonisation provides a logical treatment strategy to prevent the development of overt metastases, and thus reduce mortality.

There is evidence that only a small subset of cancer stem cells (CSCs) with an aggressive phenotype are capable of undergoing metastasis[7,8]. Little is known, however, about the microenvironmental factors aiding the colonisation of disseminated CSCs following their arrival in the bone marrow. In this work, we have utilised in vitro culture of both primary human bone and primary human breast cancer samples, along with in vivo bone metastasis models, to identify specific bone marrow-derived factors and subsequent downstream signalling in cancer cells stimulating CSC colonisation in bone.

The inflammatory cytokine IL1β has long been proposed as an important cytokine for metastasis, with recombinant IL1β first shown to promote lung metastasis in mice over 25 years ago[9]. More recently, in mouse models, IL1β has been shown to promote lung metastasis[10], and inhibition of IL1β prevent bone metastasis[11,12]. Clinically, it was observed in an atherosclerosis study that patients taking the IL1β inhibitor Canakinumab experienced reduced lung cancer incidence and mortality compared to patients taking placebo[13]. However the mechanism underpinning IL1β's contribution to metastatic growth has not yet been defined, limiting the use of inhibitors for anti-metastasis therapy. Here, we demonstrate that in the bone metastatic niche, microenvironmental IL1β promotes the ability of breast CSCs to form colonies through activation of NFKB and CREB signalling, Wnt ligand secretion and autocrine Wnt signalling in breast cancer cells. Crucially, inhibition of this pathway prevents both metastasis of breast cancer cells to bone in vivo, and CSC colony formation in the bone environment in vitro.

We propose that inhibiting IL1β-NFKB/CREB-Wnt signalling could be an important adjuvant therapeutic strategy in breast cancer to prevent overt bone metastases forming from disseminated tumour cells. Rapid translation of this approach would be possible as inhibitors to this pathway are already either licensed for other applications (Anakinra, Canakinumab, Sulfasalazine), or in clinical trials for cancer (Vantictumab, LGK974).

## Results

**Bone marrow-derived factors promote CSC colony formation.**
To investigate the effect of bone marrow-derived factors on breast CSC colony formation, bone marrow stroma from non-cancer patients was grown in culture. Bone marrow cultures were grown in continuous culture for 17 weeks (See Supplementary Fig. 1A for annotated images of bone marrow growth, showing changes in the cellular composition over time), with conditioned media removed weekly. MCF-7 breast cancer cells were treated with conditioned media from different weeks of bone marrow culture (from week 3 onwards), and CSC colony formation assessed by mammosphere formation[14,15] (See Supplementary Fig. 1B for bone marrow sample workflow). Conditioned media from bone marrow which had been cultured for 3 weeks did not stimulate CSC colony formation compared to treatment with control media (media not conditioned by the bone marrow), whereas conditioned media from bone marrow cultured for 5–17 weeks (termed CM) significantly increased CSC colony formation (Supplementary Fig. 1C). CM stimulated CSC colony formation in 5/6 breast cancer cell lines (oestrogen receptor positive (ER+): MCF-7 ($p < 0.0001$), T47D ($p < 0.0001$). oestrogen receptor negative (ER−): SUM149 ($p < 0.0001$), SUM159 ($p < 0.0001$), MDAMB231_BH ($p < 0.0001$)) (mammosphere formation and representative mammosphere images shown in Fig. 1a). Interestingly, CM did not stimulate mammosphere formation in parental MDA-MB-231 cells, but was stimulatory in a bone homing variant of this cell line (termed MDA-MB-231_BH) (MDA-MB-231_IV derived in ref. [16]). To determine if stimulation of CSC colony forming activity was specific to bone marrow stroma or could be generated by other bone cells, conditioned media from the osteoblast cell line hFOB1.19 was also assessed (oCM). This did not increase CSC colony formation compared to control treatment in MCF-7 cells (Supplementary Fig. 1D).

CSC colony formation in response to CM was next assessed in early breast cancer samples taken at the point of surgery for primary tumour removal (Fig. 1b, $n = 17$, individual patients denoted by BB6RC number. Green bars; ER positive samples, Yellow bars; ER negative samples). CM significantly increased mammosphere colony formation in 15/17 samples. This was seen in both ER+ and ER− samples (mammosphere formation for individual patients and representative mammosphere images shown in Fig. 1b), and there was no significant difference between oestrogen receptor subgroups when results from patients from each subgroup were combined (Fig. 1c). Mammosphere self-renewal (secondary mammosphere generation) was assessed in 4 of these samples (2 ER+, 2 ER−). Self-renewal was significantly increased in cells which had been treated with CM compared to control ($p = 0.049$) (Fig. 1d). These results indicate that normal bone marrow-derived factors promote the colony forming ability of breast CSCs. We also assessed the effect of CM on the ALDH+ cell population (a further CSC marker) using the Aldeflour assay[17] (Fig. 1e). CM treatment significant decreased the percentage of ALDH+ cells in the MCF-7 cell line ($p = 0.0139$), suggesting that bone marrow-secreted factors specifically enhance the CSC functional property of colony formation. We next assessed if the bone microenvironment was able to promote CSC colony formation in vivo. MCF-7 cells were grown either in vitro, or in vivo following subcutaneous or intra-femoral injection, prior to tumour dissociation and assessment of mammosphere colony formation. Cells which had grown in the femur demonstrated a 4-fold increased mammosphere formation capability ex vivo than cells grown in vitro or subcutaneously ($p = 0.0001$). This effect diminished following in vitro passage ($p = 0.0012$) (Fig. 1f). These results demonstrate that factors produced by both human and mouse bone promote the ability of breast CSCs to form colonies.

To further elucidate how bone marrow-derived factors drive metastasis, we assessed if CM could additionally promote breast cancer cell migration. When assessed by scratch assay, CM did not induce migratory ability in MCF-7 cells (Fig. 1g). Therefore, in our models, the bone microenvironment does not promote migration of breast cancer cells, instead it specifically induces breast CSCs to form colonies following arrival.

**CSC colony formation in bone marrow is mediated by Wnt.**
Next, we investigated signalling pathways promoting breast CSC colony formation in bone. Wnt signalling was identified as a strong candidate, due to both its well documented role in breast CSC maintenance and tumour initiation (reviewed in ref. [18]), and its reported role in promoting metastatic colonisation in lung[19]. We assessed if cancer cells grown in the bone microenvironment

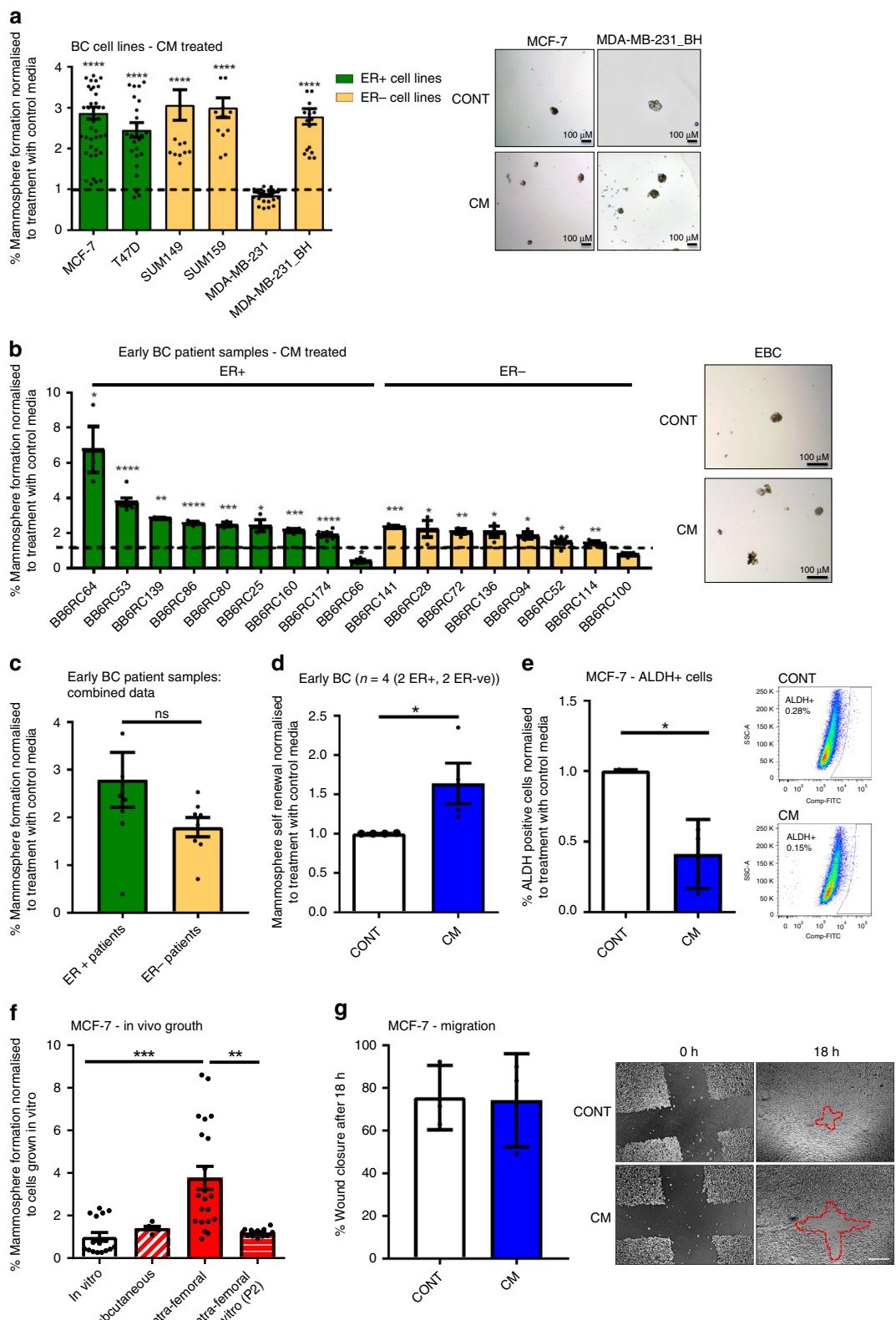

possessed increased Wnt signalling by assessing expression of the Wnt target gene *AXIN2*. *AXIN2* was increased fourfold in MCF-7 cells following CM treatment ($p = 0.016$), and 40 fold following growth in the bone ($p = 0.05$) (Fig. 2a). We then assessed the functional effect of adding recombinant Wnt3A to bone marrow cultures. Recombinant Wnt3A recapitulated the effect of CM in stimulating mammosphere colony formation in breast cancer cell

lines (MCF7; $p < 0.0001$, MDA-MB-231_BH; $p < 0.0001$) (Fig. 2b), and inhibiting Wnt using recombinant human DKK1 (inhibitor of the Wnt co-receptor LRP5[20]), or Vantictumab (therapeutic antibody to Frizzled receptors[21]) (Western blots showing reduction in protein levels of active β-catenin in Supplementary Fig. 2A), reversed the effect of CM on breast CSC colony formation in cell lines (MCF7; DKK1 $p < 0.0001$,

**Fig. 1** Bone marrow-derived factors promote breast cancer stem cell colony formation. **a** Bone marrow conditioned media (CM) from three individual patients increased mammosphere colony formation in 5/6 breast cancer cell lines (ER+; oestrogen receptor positive, ER−; oestrogen receptor negative) compared to control treatment. Representative photomicrographs of mammospheres formed by MCF-7 and MDA-MB-231_BH cells treated with control media (CONT) and CM are shown. Scale bar 100 μM. **b** CM increased mammosphere colony formation in 15/17 early breast cancer (EBC) samples. Error bars represent six technical replicates for each sample. Representative photomicrographs of mammospheres formed by one sample treated with control and CM are shown. Scale bar 100 μM. **c** Response to CM did not differ when samples were grouped by ER status ($n = 9$ ER+, 8 ER−). **d** CM increased mammosphere self-renewal in four early breast cancer samples. **e** CM treatment did not increase the percentage of ALDH+ cells in the MCF-7 cell line. Examples of representative FACS plots are shown. **f** Mammosphere formation of MCF-7 cells increased ex vivo when cells were grown intra-femorally, compared to cells grown in vitro or subcutaneously in vivo, and this increase in mammosphere formation was lost following growth of isolated cells on plastic (shown at passage 2, (P2)). **g** CM did not increase the ability of MCF-7 cells to migrate. Data is presented as **g** percentage wound closure following 18 h in culture, adjusted for wound size. Representative images of cells at 0 and 18 h shown. Scale bar 200 μM. Dashed line shows wound edge at 18 h. All graphs represent mean ± SEM, *$p < 0.05$, **$p < 0.01$, ***$p < 0.001$, ****$p < 0.0001$

Vantictumab $p = 0.0432$. MDA-MB-231_BH; DKK1 $p < 0.0001$, Vantictumab $p < 0.0001$) (Supplementary Fig. 2B), and in early breast cancer samples (DKK1; $p = 0.0222$, Vantictumab; $p = 0.0060$) ($n = 10$, Fig. 2c). Vantictumab also reversed the effect of CM on breast CSC colony formation in a further ER+ cell line (T47D, Supplementary Fig. 2C). Wnt inhibitors did not reduce mammosphere formation in MCF-7 cells treated with control media (Supplementary Fig. 2B). The specificity of this effect to Wnt signalling was confirmed in MCF-7 cells by dual gene knockdown of the LRP5 and LRP6 receptors using siRNA (Supplementary Fig. 2D, E), and by using breast cancer cells expressing a dominant negative TCF4 (EdTP from ref. [22]) (Supplementary Fig. 2F).

Next, the effect of anti-Wnt treatment on CSC colony formation was tested in vivo. Mice were injected with MCF-7 cells intra-femorally and treated systemically with Vantictumab. Mice were imaged 24 h post-injection to assess injection efficiency (Untreated control; tumours cells present in 7/10 (70%) femurs, Vantictumab; tumour cells present in 13/20 (65%) femurs) (Supplementary Fig. 2G). Mice where tumour cells were not observed 24 h post injection were excluded from further analysis. Surprisingly, no significant difference in tumour formation was observed between control and Vantictumab treated mice at experiment termination (Control; tumours present in 5/7 (71%) femurs, Vantictumab; tumours present in 11/13 (85%) femurs) (Fig. 2d), and there was no difference in tumour growth (Supplementary Fig. 2H). However, when breast cancer cells were isolated from femurs and grown as mammospheres ex vivo (representative image of mammosphere brightfield and tdTomato expression in Fig. 2e), cells from Vantictumab treated mice formed significantly less mammospheres than cells from control mice (Fig. 2e). These data demonstrate that systemic inhibition of Wnt signalling can reverse breast CSC colony formation induced by the bone environment in vivo.

**High Wnt signalling in tumours predicts poor prognosis.** Having identified Wnt signalling as a potential driver of bone metastatic colonisation, we wanted to assess if expression of this pathway in breast tumours had clinical significance. We assessed gene expression levels of Wnt pathway components in published Affymetrix datasets where metastasis data was known. We compared patient outcome between patients with absent/marginal expression of the Wnt inhibitor DKK1 (i.e. patients with high Wnt signalling tumours, 187 patients) to patients where DKK1 was present (i.e. patients with low Wnt signalling tumours, 373 patients) in 560 primary tumours from 3 published Affymetrix datasets integrated with batch correction. When tumours metastasised to bone (but not lung or brain), patients with higher tumour expression of DKK1 had a better outcome than patients whose tumours expressed lower levels of DKK1 ($p = 0.01$, log rank test of a Cox-proportional hazards model) (Supplementary

Fig. 2I). In two further datasets, we restricted the analysis to patients with only single-site bone metastases. In these patients, the association between higher tumour expression of DKK1 and better survival outcome was stronger (NKI; $p = 0.005$. Cb560; $p = 1e^{-5}$, log rank test of a Cox-proportional hazards model) (Fig. 2f). This supports our pre-clinical data suggesting an association between Wnt signalling in tumours and the ability of cancer cells to form colonies in bone.

To further explore the role of DKK1 specifically in colony formation in the bone environment, we assessed if we could induce a response to bone marrow CM by knocking down DKK1. DKK1 was knocked down by siRNA in MDA-MB-231 cells (Western blot in Supplementary Fig. 2J) which did not previously respond to CM (as shown in Fig. 1a). When DKK1 was knocked down, MDA-MB-231 cells became responsive to CM ($p < 0.0001$) (Fig. 2g). We also compared DKK1 gene expression levels in the MDA-MB-231 and MDA-MB-231_BH cell lines. The bone homing MDA-MB-231_BH cells had significantly lower expression of DKK1 than the parental cells ($p = 0.0031$) (Fig, 2h). These data add to our previous results indicating that Wnt signalling promotes the ability of CSCs to form colonies in the bone environment in vitro, and suggest a role for DKK1 specifically in promoting this, which warrants further investigation in vivo.

**Breast CSC colony formation is promoted by IL1β.** Next, we sought to identify the source of Wnt ligands driving breast CSC colony formation. We assessed expression of Wnt ligands by Affymetrix arrays in three bone marrow samples, comparing bone marrow cultured for 3 weeks (where CM did not stimulate breast CSC colony formation) to the same bone marrow cultured for 8 weeks (where CM stimulated CSC colony formation) (CSC colony formation shown in Supplementary Fig. 3A). Wnt ligand expression did not increase in stimulatory bone marrow (Fig. 3a), suggesting that the Wnt ligands promoting CSC colony formation are not produced by the bone marrow. As confirmation, Wnt ligand secretion from patient bone marrow was inhibited by treating a bone marrow sample in culture with the porcupine inhibitor (PORCNi) LGK974[23] prior to collection of CM (PORCNi inhibited Wnt signalling in a TCF4 driven Wnt reporter assay in Wnt secreting L cells (Supplementary Fig. 3B)). CM taken from bone marrow where Wnt ligand secretion had been prevented did not affect the ability of CM to stimulate CSC colony formation, confirming array results ($p < 0.0001$, PORCNi on bone marrow (BM) in Fig. 3b). Next, we tested if Wnt ligands are instead secreted by breast cancer cells, by treating MCF-7 cells with PORCNi to prevent Wnt ligand secretion, prior to treatment of these breast cancer cells with CM. PORCNi treated breast cancer cells where Wnt ligand secretion was prevented no longer responded to CM (PORCNi on MCF-7, Fig. 3b), indicating that Wnt ligands driving Wnt signalling and CSC colony formation in

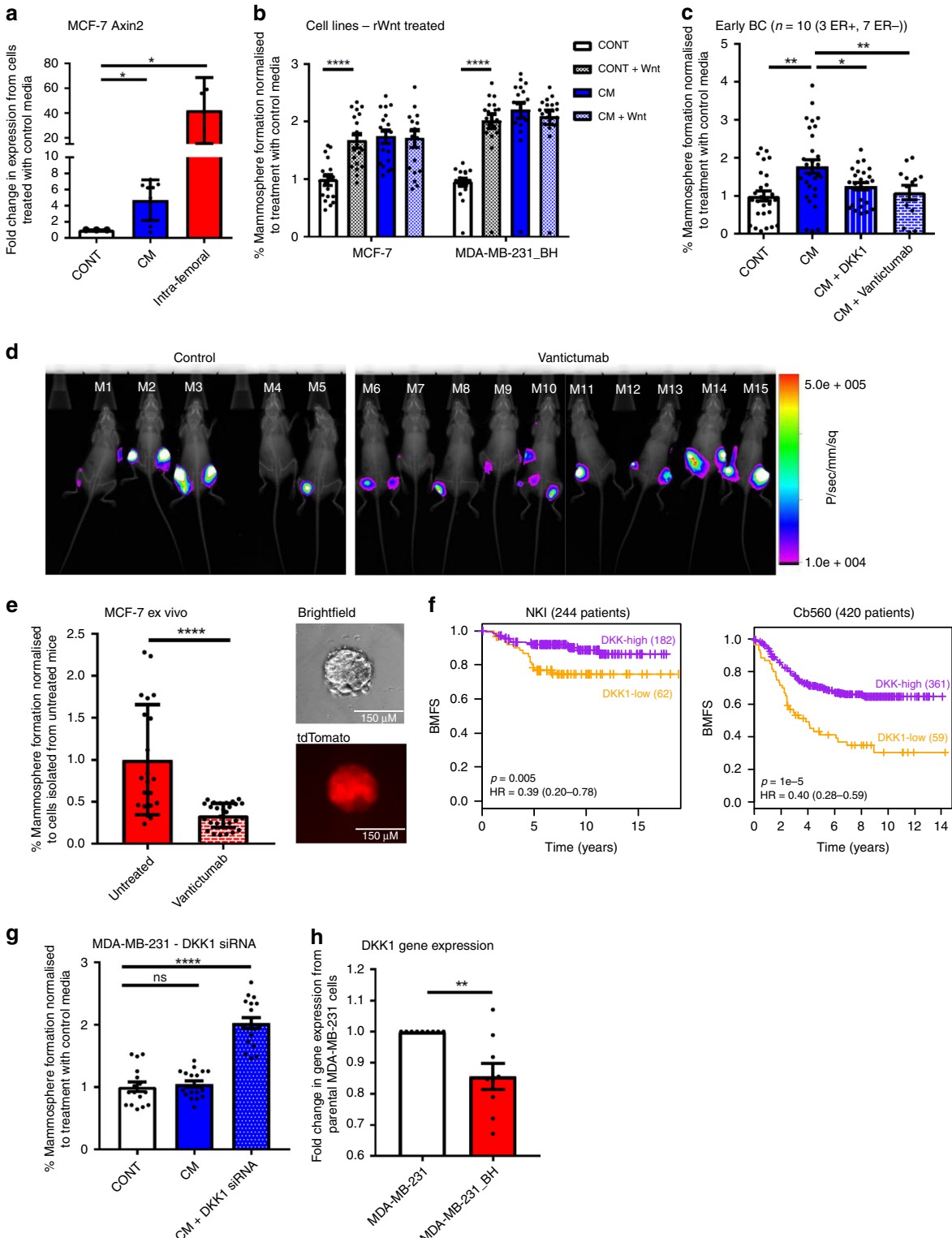

breast cancer cells are produced by cancer cells in response to intermediate bone marrow-derived factors.

To identify candidate bone marrow-derived factors driving breast CSC colony formation, cytokine arrays were performed on CM from a bone marrow sample at week 3 in culture (non-stimulatory) and week 8 in culture (stimulatory) (Fig. 3c). Several cytokines were increased in week 8 CM, with the largest fold increases in IL15 and IL1β (Blots shown in Fig. 3c. Table showing the 10 cytokines with the largest fold increases shown in

Supplementary Fig. 3C). Although IL15 and IL1β showed the largest fold increases, overall levels of these cytokines were still low, therefore the highest expressing cytokines (IL6 and IL8) were also investigated. Inhibiting IL6, IL8 and IL15 in vitro using neutralising antibodies in concurrent treatment with CM did not prevent the stimulation of CSC colony formation in MCF-7 cells by CM, however inhibition of IL1β by neutralising antibody reversed this (Fig. 3d). As IL1β has previously been implicated in bone metastasis[11], this cytokine was viewed as a strong candidate

**Fig. 2** CSC colony formation in bone marrow is mediated by Wnt signalling. **a** Treatment of MCF-7 cells with CM in vitro, or intra-femoral growth in vivo increased expression of *AXIN2* compared to cells treated with control media in vitro. **b** Addition of 100 nM recombinant Wnt (rWnt) to control media replicated the effect of CM in increasing mammosphere formation in MCF-7 and MDA-MB-231_BH cells. Adding Wnt to CM did not further increase mammosphere formation (bars labelled as CM + Wnt). **c** Inhibition of Wnt signalling with 50 ng/ml DKK1 or 50 μg/ml Vantictumab reversed induction of mammosphere formation by CM in early breast cancer samples in vitro ($n = 10$). **d** There was no significant difference in tumour formation between control and Vantictumab treated mice following intra-femoral injection of MCF-7 cells. **e** Cells isolated from femurs of Vantictumab treated mice formed significantly fewer mammsopheres ex vivo than cells isolated from control mice. Representative photomicrographs shows brightfield and tdTomato images of mammospheres. Scale bar 150 μM. **f** Survival analysis based on DKK1 expression (Low (orange) vs High (purple)) in 2 published datasets (NKI, Cb560) in patients with bone metastases. Low DKK1 expression in primary breast cancers was associated with poor outcomes in patients with bone metastasis. (NKI; significant cut points $P < 0.05$ 92/244 (38%), Cb560; significant cut points $P < 0.05$ 170/420 (40%)). HR; Hazard ratio. BMFS; Bone metastasis free survival. **g** Knockdown of DKK1 with 25 nM siRNA sensitised MDA-MB-231 cells to CM, resulting in induction of mammosphere formation. **h** DKK1 gene expression is decreased in MDA-MB-231_BH cells compared to the parental MDA-MB-231 cells. Data presented as fold change in gene expression in MDA-MB-231_BH cells compared to parental cells. All graphs represent mean ± SEM, *$p < 0.05$, **$p < 0.01$, ***$p < 0.001$, ****$p < 0.0001$

for driving breast CSC colony formation in the bone. We quantified IL1β expression in CM from four patients, comparing IL1β levels between week 3 (non-stimulatory) CM and week 8–12 (stimulatory) CM. Confirming cytokine array results, IL1β levels were low (range 4.95–8.76 pg/ml), but significantly increased in stimulatory CM ($p = 0.0025$) (Fig. 3e). Immunohistochemical analysis showed IL1β to be expressed by both mouse femur and human rib bone (Fig. 3f), demonstrating that this cytokine is present in the bone environment. This present study has utilised immune deficient mice for in vivo experiments, however we also assessed the presence of IL1β in immune competent models by ELISA analysis and immunohistochemical staining of bone sections. ELISA analysis showed significantly higher levels of IL1β in the BALB/c immune competent model compared to the NSG immune deficient model ($p = 0.01$), and this is supported by immunohistochemical staining (Supplementary Fig. 3D, E). Recombinant IL1β was added to CSC cultures in vitro, and it recapitulated the effect of CM in stimulating CSC colony formation in breast cancer cells (MCF7; $p < 0.0001$, MDA-MB-231_BH; $p < 0.0001$) (Fig. 3g).

These results identify IL1β as a bone marrow-secreted factor, however it has previously been suggested that tumour-derived IL1β promotes bone metastasis[11,12]. To test this, we inhibited IL1β by neutralising antibody in a bone marrow sample, prior to treating breast cancer cells in culture with conditioned media from this bone marrow and assessing CSC colony formation. When IL1β was inhibited in bone marrow, CM from this bone marrow no longer stimulated mammosphere formation in MCF-7 cells (IL1β NA on bone marrow (BM), Fig. 3h). Next, we inhibited IL1β by neutralising antibody in cancer cells, prior to treatment with CM. IL1β inhibition in MCF-7 cells did not affect the ability of CM to promote CSC colony formation (IL1βNA on MCF-7, $p < 0.0001$, Fig. 3h). This indicates that the IL1β driving Wnt-dependent colony formation in breast CSCs in vitro is bone marrow derived.

IL1β signals through the IL1 receptor (IL1R). We therefore sought to determine the location of this receptor in breast cancer cells. We compared IL1R expression in MCF-7 cells between bulk cells and anoikis resistant CSCs. ILR expression was increased in CSCs ($p = 0.05$) (Fig. 3i). Therefore we hypothesise that metastatic dissemination is selecting for IL1R+ CSCs that colonise the IL1β-producing bone marrow.

Finally, we tested the effect of inhibiting IL1β signalling on colony formation in the bone environment in vitro using Anakinra, an IL1 receptor antagonist. Anakinra prevented induction of CSC colony forming activity by CM in breast cancer cell lines (MCF-7; $p < 0.0001$, MDA-MD-231_BH; $p < 0.0001$, Supplementary Fig. 3F) and in patient derived samples ($n = 3$, $p < 0.0001$) (Fig. 3j). Anakinra did not reduce mammosphere formation in cells treated with control media (Supplementary

Fig. 3F). This demonstrates that CSC colony formation driven by IL1β from the bone environment can be prevented using a clinically available drug.

**IL1β promotes Wnt signalling via NFκB and CREB.** Since bone marrow-derived IL1β promotes breast CSC colony formation, and this is Wnt dependent, we assessed the ability of IL1β to activate the Wnt pathway. IL1β treatment significantly increased Wnt signalling in a 293T cell Wnt-reporter model ($p < 0.0001$) (Fig. 4a), and *Lef1* Wnt target gene expression in breast cancer cell lines (MCF7; $p < 0.0001$, MDA-MB-231_BH; $p = 0.0083$) (Fig. 4b). To determine if specific Wnt ligands are produced by breast cancer cells in response to IL1β, and to assess if any differences were seen between breast cancer subtypes, Wnt ligand gene expression was assessed in ER+ MCF-7 cells, ER− MDA-MB-231_BH cells, and patient derived samples ($n = 3$ ER+, $n = 3$ ER−) following IL1β treatment. In MCF-7 cells expression of Wnt3A, Wnt4, Wnt5A, Wnt7A and Wnt10A were significantly increased upon IL1β treatment (Wnt3A; $p = 0.015$, Wnt4; $p = 0.0028$, Wnt5A; $p = 0.0194$, Wnt7A; $p = 0.0056$, Wnt10A; $p = 0.0373$) (Fig. 4c), and in MDA-MB-231_BH cells expression of Wnt 3A and 7A were significantly increased upon IL1β treatment (Wnt3A; $p = 0.0449$, Wnt 7A; $p = 0.0139$) (Fig. 4c). In patient derived samples, increases in expression of Wnt ligands >1.5 fold were observed in both ER+ and ER− samples, with the specific Wnt ligands induced varying between samples (Fig. 4d) (3 ER+ and 3 ER− samples included, samples were either primary breast cancers (ER+ 1 and ER− 1) or metastatic cells derived from ascitic fluids (ER+ 2,3 and ER− 2,3)). Interestingly, both ER+ cell lines and patient samples appear to show a higher degree of induction of Wnt ligand expression than ER−. These data demonstrate that Wnt ligands are produced in breast cancer cell lines and patient samples in response to IL1β, and the Wnt ligands driving CSC metastatic colony formation may vary between patients.

To determine the intra-cellular signalling mechanism by which IL1β promotes Wnt signalling, dual-luciferase pathway reporter arrays were performed on MCF-7 cells following IL1β treatment. Luciferase reporting for the NFKB and CREB pathways were increased in MCF-7 cells (Fig. 4e), and an increase in phosphorylation of CREB and nuclear NFKB following IL1β treatment was confirmed in both MCF7 and MDA-MB-231_BH cells (Fig. 4f, g). When NFKB signalling was inhibited using Sulfasalazine[24], or CREB signalling was inhibited using KG-501[25], there was a significant reduction in IL1β-induced Wnt reporting in 293T cells (Fig. 4h), and a significant reduction in IL1β-induced expression of *Lef1* in both MCF-7 (Sulfasalazine; $p = 0.0003$, KG-501; $p = 0.0084$) and MDA-MB-231_BH cells (Sulfasalazine; $p = 0.0045$, KG-501 $p = 0.0007$) (Fig. 4i). Treatment with Sulfasalazine significantly reduced IL1β-induced

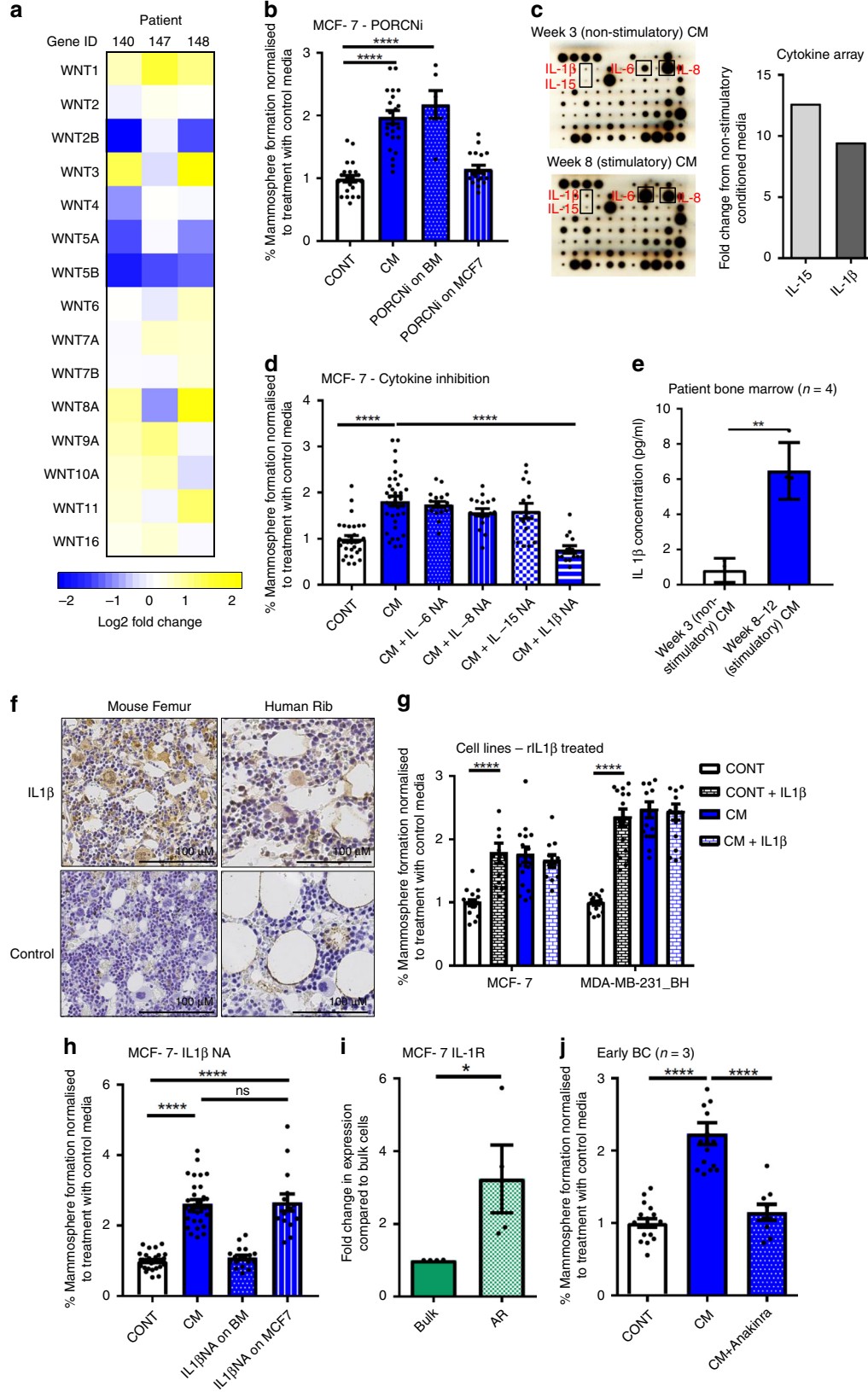

expression of Wnt3A ($p = 0.0045$), Wnt4 ($p = 0.0003$) and Wnt7A ($p < 0.0001$), and treatment with KG-501 significantly reduced IL1β-induced expression of Wnt3A ($p = 0.0022$), Wnt4 ($p = 0.0141$), Wnt5A ($p < 0.0001$) and Wnt7A ($p < 0.0001$) in MCF-7 cells (Fig. 4j). These findings establish CREB and NKFB

as important cellular pathways downstream of IL1β and upstream of Wnt secretion and signalling. We also tested the effect of inhibiting NFKB and CREB signalling on CSC colony formation in vitro. Inhibiting NFKB and CREB reversed the effect of CM in cell lines (MCF7; sulfasalazine $p < 0.0001$, KG-501 $p < 0.0001$.

**Fig. 3** Wnt-dependent breast CSC colony formation is promoted by IL1β. **a** Gene expression levels of Wnt ligands did not increase from week 3 to week 8 bone marrow taken from three patients (patient numbers: 140, 147, 148). Data is presented as log2 fold change in gene expression from week 3 to week 8. **b** Treating bone marrow in culture with 100 μM porcupine inhibitor (PORCNi) LGK974 for 72 h prior to taking CM did not prevent CM stimulating mammosphere formation in MCF-7 cells (PORCNi on BM). Treating MCF-7 cells with 100 μM of LGK974 for 72 h prior to the addition of CM prevented stimulation in mammosphere formation (PORCNi on MCF7). **c** IL15 and IL1β showed the largest increase from week 3 to week 8 CM when analysed by cytokine array ($n = 1$). **d** Inhibition of IL6, IL8 or IL15 with 5 μg/ml neutralising antibody did not prevent induction of mammosphere formation by CM in MCF-7 cells, whereas inhibition of IL1β with 5 μg/ml neutralising antibody prevented CM from increasing mammosphere formation. **e** IL1β levels were significantly increased in week 8–12 stimulatory CM compared to week 3 non-stimulatory CM ($n = 4$). **f** IL1β is expressed by both normal mouse and human bone. Staining is representative of three sections from three independent samples. Femur taken from 8 week old NSG mice. Negative controls (no primary antibody) are shown. Scale bar 100 μM. **g** The addition of 10 ng/ml recombinant IL1β to control media replicated the effect of CM in stimulating mammosphere formation in MCF-7 and MDA-MB-231_BH. **h** Treating a bone marrow sample in culture with 5 μg/ml of IL1β neutralising antibody for 72 h prior to collection of CM prevented stimulation of mammosphere formation in MCF-7 cells (IL1βNA on BM). Treating MCF-7 cells with 5 μg/ml of IL1β neutralising antibody for 72 h prior to the addition of CM did not prevent stimulation in mammosphere formation by CM (IL1βNA on MCF7). **i** IL1R gene expression is increased in the anoikis resistant (AR) cancer stem cell population in MCF-7 cells. **j** 10 μg/ml Anakinra reversed the induction in mammosphere formation by CM in early breast cancer samples ($n = 3$). All graphs represent mean ± SEM, $*p < 0.05$, $**p < 0.01$, $***p < 0.001$, $****p < 0.0001$

MDA-MB-231_BH; sulfasalazine $p < 0.0001$, KG-501 $p < 0.0001$) (Supplementary Fig. 4) and patient derived samples ($n = 3$, sulfasalazine; $p = 0.0004$, KG-501; $p < 0.0001$) (Fig. 4k). In cell lines, combined treatment with both sulfasalazine and KG501 further reduced mammosphere formation compared to either treatment alone (MCF7; CM + sulfasalazine vs CM + Sulfasalazine + KG501 $p < 0.0001$, CM + KG501 vs CM + Sulfasalazine + KG501 $p = 0.0020$. MDA-MB-231_BH; CM + sulfasalazine vs CM + Sulfasalazine + KG501 $p < 0.0001$, CM + KG501 vs CM + Sulfasalazine + KG501 $p < 0.0001$) (Supplementary Fig. 4) .Taken together, our data demonstrate that IL1β stimulates Wnt-dependent CSC colony formation in the bone environment via an induction of NFκB and CREB signalling in breast cancer cells.

**Inhibition of IL1β-Wnt signalling prevents bone metastasis**. Finally, we assessed if inhibiting microenvironmental IL1β or tumour cell Wnt signalling could prevent bone metastases in vivo. We initially used MDA-MB-231_BH cells as these are bone homing following tail vein injection[16]. All mice were injected with tumour cells, and mice were treated with an anti-mouse IL1β antibody, the anti-human Frizzled receptor antibody Vantictumab, or with vehicle control. Tumour formation was monitored by bioluminescence, and assessed and quantified by histological examination of bone tissue sections from all hind limbs. We also assessed the effect of IL1β and Wnt inhibition on the bone environment by micro-CT. Metastatic breast cancer patients often experience bone loss as part of their disease progression, therefore any treatments which promote bone volume could be advantageous for these patients.

IL1β inhibition reduced tumour formation from 11/16 hind limbs in control mice to 3/16 hind limbs in treated mice (69% vs. 19%, $p = 0.0113$, chi squared test) (Representative luciferase images of whole mice shown in Supplementary Fig. 5A, excised limbs in Supplementary Fig. 5B, representative H&E stained sections in Fig. 5a). Anti-IL1β treatment also significantly increased trabecular bone volume compared to control mice (representative images and graphical representation of 8 scored femurs per treatment group in Fig. 5b). Systemic Wnt inhibition reduced tumour formation from 9/16 hind limbs of control mice to 1/14 hind limbs in treated mice (56% vs 7%, $p = 0.0067$, chi squared test) (Representative luciferase images of whole mice shown in Supplementary Fig. 5C, LightTools images of excised hind limbs in Supplementary Fig. 5D, representative H&E stained sections in Fig. 5c). Wnt inhibition appeared to increase trabecular bone volume compared to untreated tumour-bearing mice (%BV/TV $p = 0.0034$, graphical representation in Supplementary Fig. 5E), however this was due to a reduction in total bone volume (representative bone images in Fig. 5d, arrows

denote areas of bone loss; analysis of cortical bone volume in Supplementary Fig. 5F showing reduction in cortical bone volume in Vantictumab treated mice ($p < 0.0001$)) rather than a bone building effect.

Our in vitro data suggests that Wnt inhibition does not prevent tumour migration, but instead prevents colony formation in the bone. To confirm this in vivo, two-photon analysis of injected DiD labelled tumour cells was performed to assess if tumour cells were present in the bone of mice treated with Vantictumab, despite less overt tumours developing in this group. Twenty-eight days following injection, similar numbers of tumour cells were present in the proximal tibiae of control or Vantictumab treated mice (Fig. 5e), demonstrating that inhibiting Wnt signalling does not prevent tumour cells homing to bone.

Given that IL1β inhibition prevented tumour colonisation in bone whilst promoting trabecular bone volume, we also assessed the ability of IL1β inhibition to prevent metastatic colonisation in a spontaneously bone metastatic ER + PR + Her2− patient derived xenograft (PDX) model (BB3RC32 from ref. [26]) following intra-cardiac injection. IL1β was inhibited through systemic treatment with Anakinra, and tumour formation assessed as previously. IL1β inhibition reduced tumour formation from 10/24 hind limb tumours in control mice to 3/22 hind limb tumours in Anakinra treated mice (42% vs 14%, $p = 0.021$, chi squared test) (Representative images of excised hind limbs in Supplementary Fig. 5G, representative H&E images in Fig. 5f). As we had previously observed in our cell line model that tumour cells were present in the bones of treated mice which did not develop overt metastases, we also assessed this in this model. Tumour cells were present in Anakinra treated mice which had not developed metastases at experiment termination (Supplementary Fig. 5H), suggesting that, as in the cell line model, inhibiting this signalling pathway prevents colonisation of disseminated tumour cells in bone, rather than tumour cell homing to bone. Taken together, these PDX data demonstrate that in this more clinically relevant model, systemic inhibition of IL1β using a clinically available drug prevents metastatic colonisation. Our in vivo data strongly support in vitro results, and demonstrate that targeting IL1β-Wnt signalling can prevent disseminated CSCs from forming overt bone metastases. The signalling mechanism identified in this study, and drugs tested which could be utilised to inhibit it clinically, are shown in Fig. 5g.

## Discussion
In this work we establish IL1β-NFKB/CREB-Wnt as a novel signalling pathway regulating the metastatic colonisation of disseminated breast CSCs in the bone microenvironment.

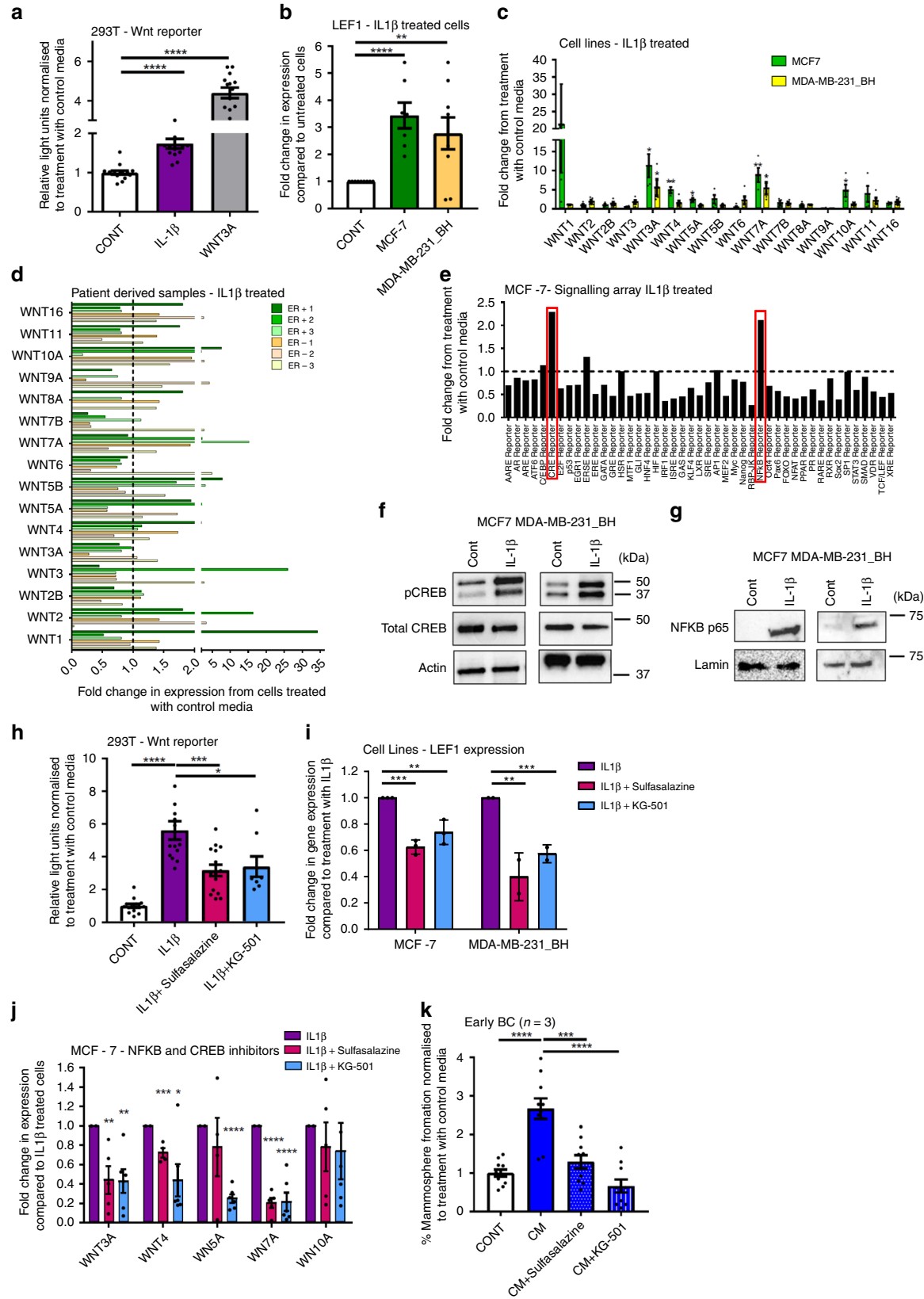

Using primary tissue, we have shown for the first time that normal human bone marrow-derived factors enhance the ability of breast CSCs to form metastatic colonies. Interestingly, although we saw an increase in breast CSC colony formation in response to the bone environment, this was not accompanied by

an increase in ALDH+ cell number. It has previously been reported that there are multiple CSC populations within tumours with different functional properties[27], and our data support this. Of particular clinical interest is the finding that there is no difference in CSC colony forming response to bone marrow-derived

**Fig. 4** IL1β promotes Wnt signalling in breast cancer cells via an induction in NFKB and CREB. **a** IL1β treatment stimulated Wnt reporting in a 293T 7TCF cell line. 100 nM Wnt3A treatment included as a positive control. **b** IL1β treatment increased expression of Lef1 in both MCF-7 and MDA-MB-231_BH cells compared to treatment with control media. **c** IL1β treatment increased gene expression levels of Wnt ligands Wnt3A, 4, 5A, 7A and 10A in MCF-7 cells and Wnt3A and 7A in MDA-MB-231_BH cells. **d** IL1β treatment increased gene expression levels of some Wnt ligands in patient derived samples (n = 6). ER positive samples; green bars, ER negative samples; orange bars. Dashed line set as fold change of 1. Note split axis to show lower fold changes. **e** IL1β treatment increased NFKB and CRE signalling in MCF-7 cells. **f** IL1β treatment increased phospho CREB protein, but not total CREB protein in MCF-7 and MDA-MB-231_BH cells. Actin was used as a loading control. **g** IL1β treatment increased nuclear phospho NFKB p65 protein in MCF-7 and MDA-MB-231_BH cells. Lamin was used as a nuclear loading control. **h** Treatment with sulfasalazine or KG-501 reversed the induction of Wnt signalling by IL1β in a 293T 7TCF reporting assay. **i** Treatment with sulfasalazine or KG-501 reversed the induction of Lef1 gene expression by IL1β treatment in both MCF-7 and MDA-MB-231_BH cells. **j** Treatment with sulfasalazine or KG-501 reversed the induction in gene expression of some Wnt ligands by IL1β . **k** Sulfasalazine or KG501 treatment reversed the induction in mammosphere formation by CM in early breast cancer patient derived samples (n = 3). Data presented as fold change in percentage mammosphere formation normalised to cells treated with control media. All graphs represent mean ± SEM, *p < 0.05, **p < 0.01, ***p < 0.001, ****p < 0.0001

factors between ER+ and ER− samples, despite the majority of breast cancers which metastasise to the bone being ER+[28]. Based on this finding, we speculate that the clinical difference between subtypes could be due to a difference in pre-metastatic niche formation by ER+ samples, differential homing ability, or a difference in Wnt ligand expression in response to IL1β.

We have identified IL1β as a key cytokine produced by human bone marrow which enhances the ability of breast CSCs to form colonies. IL1β has long been proposed as an important cytokine for metastasis, both in breast and other cancers[9–12]. A metastasis promoting role for IL1β is not totally clear however, and contradictory recent data has also suggested that IL1β prevents cell proliferation[29]. Our study adds further weight for a promoting role of IL1β in metastasis, and challenges the assumption that IL1β is tumour derived. Having established a role for microenvironmental IL1β in promoting metastasis, next it will be important to determine specific cells within the bone microenvironment producing this cytokine. Previous studies have shown that multiple cell types within the bone marrow produce IL1β, including both immune cells and non-immune cells[30–32].

Mechanistically, we demonstrate that bone marrow-derived IL1β promotes metastatic colony formation through activation of intracellular NFKB/CREB signalling and Wnt ligand production by cancer cells. Wnt signalling has been shown to promote metastatic colonisation in the lung[19], but this role downstream of IL1β in bone is novel. Interestingly, a recent study suggests that tumour-derived Wnt promotes IL1β secretion by macrophages, suggesting the possibility of a feed-forward signalling mechanism promoting metastasis[33]. Elucidation of this signalling pathway has identified multiple drug targets for clinical use. Canakinumab and Anakinra that inhibit IL1β signalling, and the NFKB inhibitor Sulfasalazine, are in clinical usage for other indications[24,34–36]. Vantictumab is in phase 1b clinical trials in patients with advanced cancer, and the porcupine inhibitor LGK974 is being tested in Phase I clinical trials[37]. It has been reported that Vantictumab has adverse effects on patient bone in clinical trials[13], however a clinical strategy of reduced dosing and co-administration of zoledronic acid mitigates adverse events[18], rendering this drug suitable for patient use. In our intra-femoral model, anti-Wnt treatment did not alter tumour take in bone or tumour growth, despite demonstrating an effect of CSC colony forming activity ex vivo. Vantictumab has previously been shown to inhibit tumour growth following subcutaneous injection[21], and we predict the lack of effect seen in our study may either have been due to the large number of cells injected (and a lack of selection for CSCs), or the nature of the model, where tumour cells are injected directly into the bone.

It should be noted that there are limitations to the models we have used within this study. It has been shown that tumour promoting pre-metastatic niches are established at distant sites before the arrival of disseminated cells[38,39], and since our model uses bone marrow from non-cancer patients, it fails to address this. In vivo, no single model has been identified which accurately recapitulates the multiple steps of human breast cancer metastasis, and we have selected models to allow us to best study the final colonisation step. A further limitation of our models is the use of immune-compromised mice. Whilst these are essential to study metastasis of human tumour cells in vivo, they do not give us any insight into the interplay between IL1β-Wnt inhibition and the immune system during metastatic colonisation. However, a recently published paper has shown that inhibiting IL1β in mouse models enhances tumour cell immunity by synergising with anti-PD-1[40], which is very promising for future combination therapies.

A question this study does not answer is why only some patients experience colonisation of disseminated tumour cells in bone. Thirty percent of breast cancer patients have DTCs in the bone marrow at the time of diagnosis[4], yet many of these will never develop overt metastases. Based on our data, we hypothesise that this could be due to either particular properties of the primary tumour, or a change in the bone microenvironment in individual patients. We have preliminary data showing that tumour cell-conditioned medium induces IL1β production by the bone marrow (Supplementary Fig. 6). This indicates one mechanism by which an increase in bone marrow IL1β has the potential to promote metastatic colonisation in breast cancer patients, but is a point for further investigation.

In summary, we have identified IL1β-NFKB/CREB-Wnt signalling as a novel pathway promoting the colonisation of breast CSCs in the bone microenvironment, and we have demonstrated that drugs targeting this pathway prevent both bone metastases in vivo and colony formation by breast CSCs in vitro. This work therefore provides a strong rationale for consideration of inhibitors of this pathway as an adjuvant therapeutic strategy in breast cancer to prevent metastatic tumour cell colonisation in bone, subject to confirmation of our results in immune competent mouse models.

## Methods
**Compliance with ethical standards**. All patients underwent fully informed consent in accordance with local ethics committee guidelines. Ethical approval for metastatic samples was granted by the Central Office for Research Ethics Committee study number #05/Q1403/159. Early breast cancer samples were collected via the MCRC Biobank which is licensed by the Human Tissue Authority (licence number: 30004) and approved as a research tissue bank by the South Manchester Research Ethics Committee (Ref: 07/H1003/161 + 5). All in vivo experiments were performed in accordance with the UK Home Office Animals (Scientific Procedures) Act 1986. The ethical compliance and study protocol were approved by the CRUK Manchester Institute Animal Welfare and Ethical Research Board (AWERB) and the UK Home Office. In vivo experiments utilising MDA-MB-231_BH cells and intra-cardiac injection of PDX cells were carried out at the University of Sheffield with Home Office approval under licence 70/8964. In vivo experiments utilising intra-femoral and subcutaneous injections of MCF-7 cells

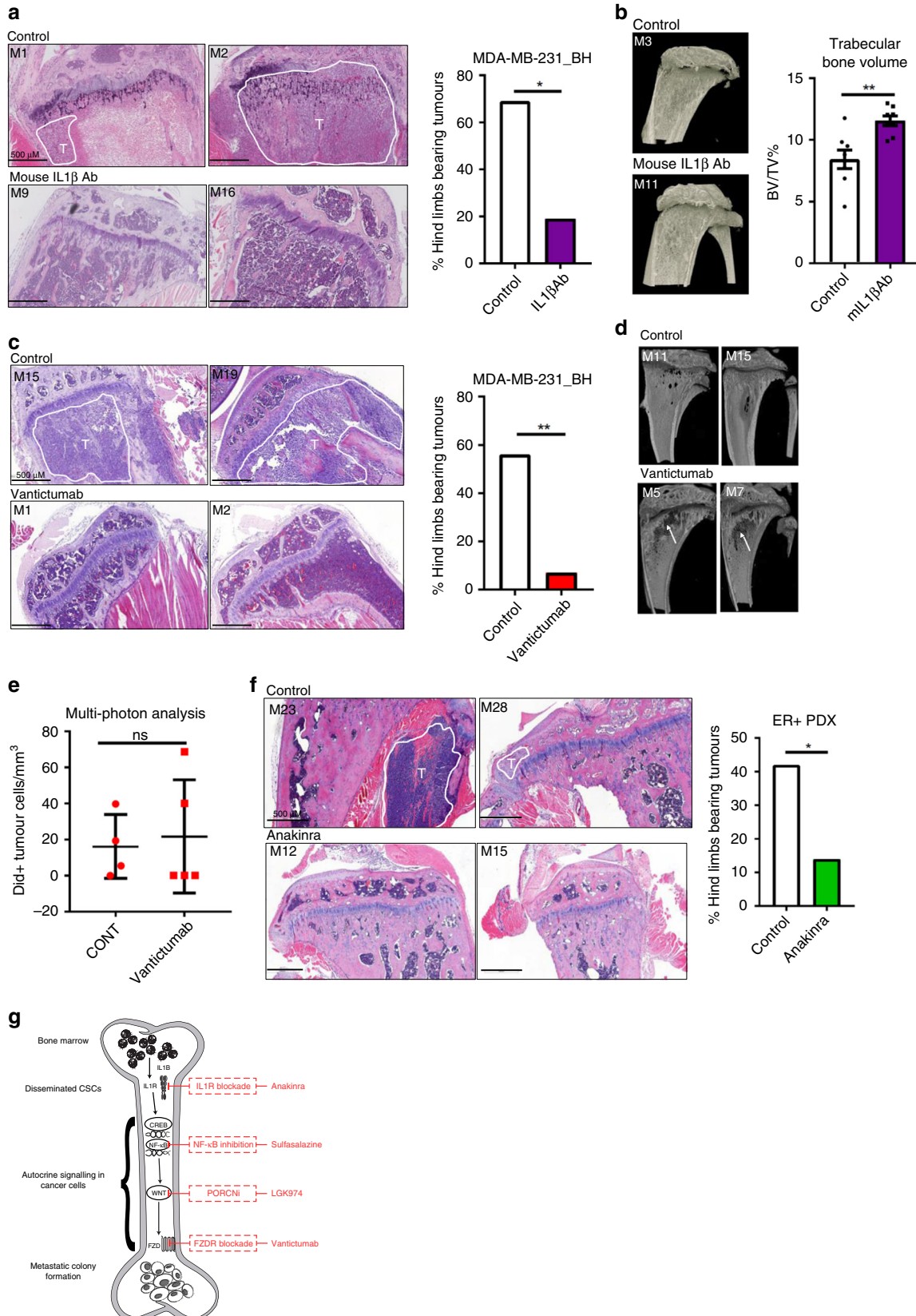

and were carried out at the University of Manchester with Home Office approval under licence 40/3645.

**Cell lines**. Cell lines were purchased from ATTC (MCF-7 (catalogue number ATCC® HTB-22), T47D (ATCC® HTB-133), 293T (ATCC® CRL-1573), MDA-

MB-231 (ATCC® HTB-26). Wnt secreting L cells were a kind gift from Professor Keith Brennan, University of Manchester. hFOB1.19 cells were a kind gift from Dr Suzanne Johnson, University of Manchester. SUM149 and SUM159 cells were a kind gift from Professor Göran Landberg, University of Manchester. All cell lines were authenticated by multiplex PCR assay (Life Technologies, Paisley, UK) and verified as mycoplasma free on a monthly basis. MCF-7, T47D, 293T, SUM149,

**Fig. 5** Systemic inhibition of either IL1β or Wnt signalling prevents bone metastasis in vivo. **a** Anti-mouse IL1β treatment reduced hind limb bone metastases in a spontaneous MDA-MB-231_BH mouse model. Histology shows tumour cells in bones of control mice (M1, M2) vs. tumour free hind limbs in treated mice (M9, M16). Tumour areas are marked as T. Scale bar 500 μM. Statistical significance assessed by chi squared test, tumour vs. no tumour in treated vs. untreated mice. **b** Anti-IL1β treatment increased trabecular bone volume at experiment termination. Example images presented from a control mouse (M3) and a mouse treated with IL1β antibody (M11). Data presented as percentage bone volume/total volume (%BV/TV) ($n = 8$). **c** Vanticutumab treatment reduced spontaneous hind limb bone metastases in the MDA-MB-231_BH model. Histology shows tumour cells in bones of control mice (M15, M19) vs. tumour free hind limbs in treated mice (M1, M2). Tumour areas are marked as T. Scale bar 500 μM. Statistical significance assessed by chi squared test, tumour vs. no tumour in treated vs. untreated mice. **d** Vanticutumab treated mice showed a thinning of cortical and spongy bone. Representative images of control mice (M11, M15) and Vanticutumab treated mice (M5, M7) are presented. Arrows denote areas of bone loss. **e** No significant difference in the number of DiD labelled tumour cells in control and Vanticutumab treated mice was seen 30 days following injection of tumour cells ($n = 4$ control, $n = 5$ Vanticutumab treated). **f** Anakinra reduced hind limb tumours in an ER + PDX (BB3RC32) model following intra-cardiac injection of tumour cells. Histology shows tumour cells in bones of control mice (M23, M28) vs. tumour free hind limbs in treated mice (M12, M15). Tumour areas are marked as T. Scale bar 500 μM. Statistical significance was assessed by chi squared test, tumour vs. no tumour in treated vs. untreated mice. **g** Our proposed model of signalling pathways underpinning bone colonisation by metastatic breast cancer cells. All graphs represent mean ± SEM, $*p < 0.05$, $**p < 0.01$, $***p < 0.001$, $****p < 0.0001$

SUM159 and Wnt secreting L cells were grown adherently in DMEM complete medium (DMEM/10% foetal bovine serum/2mM L-glutamine). MDA-MB-231 and MDA-MB-231_BH cells (bone homing variants of MDA-MB-231, originally named MDA-MB-231_IV when derived in ref. [16]) were grown adherently in RPMI complete media (RPMI/10% foetal bovine serum/2mM L-glutamine). hFOB1.19 cells were grown in DMEMF12 (10% FBS, 0.3 mg/ml G418). Cells were grown to passage 30 before discarding.

**Bone marrow culture**. Bone marrow samples were extracted from patients undergoing radical nephrectomy for non- malignant renal failure. Bone marrow samples were cultured at a density of $2 \times 10^6$ ml$^{-1}$ in long-term culture medium (Iscove's Modified Dulbecco's Medium at 350 mOsm containing 10% FCS, 10% horse serum and 50 μM hydrocortisone—referred to as control media or CONT in experiments) on plastic at 33 °C in 5% $CO_2$ in air for 20 weeks without passage. Fifty percent of bone marrow culture media was removed and replenished weekly to sustain bone marrow growth[41]. This conditioned media (referred to as bone marrow conditioned media or CM) was used in experiments. For donor characteristics see Supplementary Table 1.

**Breast cancer patient samples**. Early breast cancer samples were collected from patients undergoing primary tumour removal at three NHS trusts in Greater Manchester, UK. Early breast cancers were disaggregated and digested in 4.7 ml RPMI plus enzymes from the Miltenyi tumour dissociation kit (130-095-929) as per manufacturer's instructions for 2 h. Cells were strained through a 70 μM filter (BD, 352340) followed by a 40 μM filter (BD, 352340) and centrifuged at $1000 \times g$ for 5 min at room temperature to pellet cells[26] prior to mammosphere culture. Clinico-pathological details are presented in Supplementary Table 2. Metastatic samples were collected from palliative pleural or ascitic drainage procedures at The Christie NHS Foundation Trust. Metastatic samples were first centrifuged at $1000 \times g$ for 10 min to pellet cells. Pellets were resuspended in PBS and blood cells were removed by centrifugation of the cell suspension through 0.5 volumes of Lymphoprep solution (Axis Shield, Dundee, UK) at $800 \times g$ for 20 min[26], prior to mammosphere culture. Clinico-pathological details are listed in Supplementary Table 3. 19/24 patient samples used were treatment naïve at the time of surgery. Where patient derived samples cells were grown in culture this was overnight in DMEM F12 Glutamax supplemented with 10% FBS, 10 μg/ml insulin, 10 μg/ml hydrocortisone and 5 μg/ml EGF.

**PDX samples**. PDX models were created by implanting breast cancer samples into female NSG (NOD.Cg-Prkdcscid Il2rgtm1Wjl/SzJ) mice in accordance with the UK Home Office Animals (Scientific Procedures) Act 1986. Early breast cancers were implanted as $2 \times 2$ mm$^3$ fragments, and metastatic samples were injected as $1 \times 10^6$ cancer cells. Oestrogen supplementation in drinking water was provided for mice with ER positive tumours at a concentration of 8 μg/ml. Tumour growth was measured twice weekly using callipers. When tumours reached 1.3 cm$^3$ mice were culled and tissue fragments were digested as for early breast cancer patient samples above[26]. Where cells derived from PDX tumours were grown in culture this was overnight in DMEM F12 Glutamax supplemented with 10% FBS, 10 μg/ml insulin, 10 μg/ml hydrocortisone and 5 μg/ml EGF.

**Cell treatments**. Cell lines and patient derived samples were treated with the following prior to downstream assays: 50 ng/ml DKK1 (GF170, Milipore), 50 μg/ml Vanticutumab (Oncomed), 100 nm Wnt3A (5036-WN, R&D systems), 100 μM LGK974 (S7143, Selleckchem), 10 ng/ml rIL15 (247-ILB, R&D systems), 10 ng/ml rIL1β (201-LB, R&D systems), 5 μg/ml IL1β neutralising antibody (MAB201, R&D systems), 5 μg/ml IL15 neutralising antibody (MAB-274, R&D systems), 5 μg/ml IL6 neutralising antibody (MAB2061, R&D systems), 5 μg/ml IL8 neutralising

antibody (MAB208, R&D systems), 10 μg/ml Anakinra (Amgen, Cambridge, UK), 5 mM Sulfasalazine (Sigma), 10uM KG-501 (Sigma). Treatment times for individual assays are detailed below.

**Mammosphere assay**. For mammosphere culture, a single cell suspension was prepared by manual disaggregation of cells through a 25 gauge needle, and a total of 500 cells/cm$^2$ were plated in appropriate polyHEMA (Poly (2-hydroxyethylmethacrylate)) coated tissue culture plates in mammosphere medium (DMEM-F12/B27/20 ng/ml EGF/Pen-Strep)[15]. To assess the effect of bone marrow conditioned media on mammosphere formation, cell lines were treated in monolayer with conditioned media or control media (+/− inhibitors) for 72 h prior to plating in mammosphere culture. Patient samples cells were plated directly into a 50:50 ratio of mammosphere media and bone marrow conditioned media or control media (+/− inhibitors). Cells were cultured for five days (cell lines) or seven days (primary samples) before mammospheres greater than 50 μm were counted. Mammosphere-forming efficiency (MFE) was calculated by dividing the number of mammospheres formed by the number of cells plated and expressed as a percentage. Mammosphere data is presented as percentage mammosphere formation in treated samples, normalised to percentage mammosphere formation in control samples. For all cell line experiments, each experiment represents six technical replicates and three biological repeats. For patient sample experiments, each experiment was carried out with as many technical replicates as possible given the number of cells extracted from the sample (a minimum of three, and up to six technical replicates) and the number of biological repeats (the number of patients tested) is included in each figure. Mammosphere self-renewal was assessed when the number of mammospheres grown in primary culture was sufficient for onward passage. To assess self-renewal, mammospheres were counted, centrifuged ($115 \times g$), and dissociated into a single cell suspension by incubation for 2 min at 37 °C in trypsin EDTA 0.125% (Sigma), followed by mechanical dissociation through a 25 gauge needle[15]. No additional treatments were added in secondary generation. Mammosphere self-renewal was calculated by dividing the number of secondary mammospheres formed by the number of primary mammospheres formed.

**Aldefluor assay**. The Aldefluor assay (Stemcell Technologies) was performed according to manufacturer's instructions. 7-aminoactinomycin D (7AAD, BD) was used to exclude dead cells. Data were acquired on a LSR II (BD) flow cytometer and analysed using the BD FACSDiva™ software.

**Scratch assay**. MCF-7 cells were plated on tissue culture plates to create a confluent monolayer. A straight line was scraped with a p200 pipette tip and debris removed by washing cells once with growth media. Cells were incubated for 18 h in either control media or CM, and images taken under a brightfield microscope.

**RNA and real-time PCR**. RNA was prepared from human bone marrow samples using the Qiagen RNEasy Micro kit, and from breast cancer cells using the Qiagen RNEasy Mini kit, as per manufacturer's instructions. RNA concentration and purity was determined using an ND-1000 spectrophotometer (NanoDrop Technologies). For RT-PCR 1 μg of RNA was reversed transcribed using the TaqMan Reverse Transcription Reagents (Applied Biosystems, N8080234) as per manufacturer's instructions. Quantitative real-time PCR reactions were performed on a QS3 PCR machine (Applied Biosystems) using TaqMan® Universal PCR Master Mix (Applied Biosystems) and Taqman predesigned gene expression assays (LEF1: hs01547250_m1, 18S: hs99999901-s1, AXIN2: hs00610344_m1, GAPDH: hs03929097_g1, ACTB: hs99999903_m1) (Thermo Fisher). CT values were normalised to the average CT value of housekeeping genes (β-actin, GAPDH, 18S) and expression levels were calculated using the ΔΔCt method. For Wnt PCR arrays RNA was extracted using the Qiagen RNEasy Mini kit. RNA was reversed

transcribed using the Qiagen RT² First Strand kit as per manufacturer's instructions, followed by the Human WNT signalling pathway PCR array as per manufacturer's instructions. CT values were normalised to the average CT value of housekeeping genes (β-actin, B2M, GAPDH, HPRT1 and RPLPO) and expression levels were calculated using the ΔΔCt method.

**Microarray analysis**. Total RNA was quality assessed by Agilent Bioanalyzer. cDNA was hybridised to Human U133 2.0 Plus GeneChips (Affymetrix, High Wycombe, UK) according to the GeneChip® Expression Analysis Technical Manual (Affymetrix, Rev. 5). Probe set mapping was performed using RMA (Robust Multichip Average) Express[42]. Arrays were background corrected and Quantile normalised before the data was Probe Level Model summarised. Chip pseudo-images were then assessed for QC purposes before assessment of RLE (relative log expression) median values. RMA expression values were produced and expressed in the log2 scale. Expression levels were then compared using log2 fold changes and visualised by colour plot.

**Lentiviruses generation**. The following lentiviral plasmids were used: 7xTcf-FFluc//SV40 (as detailed in ref. [22]), pFULT (bicistronic tdTomato and luciferase expressing vector[43]), EdTP[22]. Lentivirus was prepared using the Lenti-Pac FIV expression Packaging Kit (GeneCopea) as per manufacturer's instructions. 293T cells were infected with lentivirus in transfection medium (DMEM/5% Heat Inactivated foetal bovine serum/L-glutamine) with 8 µg/ml polybrene. Media was changed to general culture medium (DMEM/10% foetal bovine serum/L-glutamine) 24 h post infection and infected cells were selected with 5 µg/ml puromycin from 48 h post infection.

**Wnt transcriptional assay**. 293T 7xTcf-FFluc//SV40 infected cells were treated with 100 nm recombinant Wnt3A or 10 ng/ml rIL1β +/− 5 mM Sulfasalazine or 10uM KG-501) for 24 h in serum free media prior to lysis with 1× Passive lysis buffer (5×, Promega, E 1941), rocked for 15 min and luminescence assessed using the Dual-Glo Luciferase assay system (Promega, E2920) following manufacturer's instructions. Luciferase activity was measured using a luminometer (Promega, Glomax Multi+ Detection System with Instinct Software).

**siRNA knockdown of LRP5, LRP6 and DKK1**. ON-TARGET Plus siRNA to LRP5, LRP6 and DKK1 (Dharmacon) was used to transfect cells using Dharmafect (Dharmacon) according to manufacturer's instructions in serum free media. All siRNA were used at a concentration of 25 nM. Media was changed after 24 h, and cells were incubated for either 48 h (DKK1) or 72 h (LRP5, LRP6) prior to use.

**Western blotting**. A total of 40 µg was utilised for each sample. Primary antibodies for Western blotting were: LRP5 (1:1000, Cell signalling 3889), LRP6 (1:1000, Abcam ab75358), Non-phospho active β-catenin (1:1000, Cell signalling 8814), DKK1 (1:100, Cell Signalling 4687), NFKB p65 (1:1000 8242P, Cell Signalling), phospho-CREB Ser133 (1:1000, 9198S). Membranes were incubated with primary antibody at 4 ⁰C overnight with gentle shaking. Immunoreactive proteins were detected by enhanced chemiluminescence (Pierce). Uncropped versions of key blots are included in the Source Data files.

**Cytokine experiments**. Cytokines present in CM were assessed using RayBio Human Cytokine Array 5 (R&D systems) according to manufacturer's instructions. MacBiophotonics Image J (1.42l) was used to quantitate cytokine spots as per manufacturer's instructions. Raw densitometry data was extracted by identifying a single exposure with a high signal to noise ratio, measuring the density of each spot using circles of equal size dimensions, and determining the summed signal density across the entire circle for each spot. Background signal was then subtracted and data was normalised based on positive control signals for each array.

**ELISA**. IL1β was detected in human bone marrow conditioned media using the Human IL1β ELISA MAX™ Deluxe ELISA kit (Biolegend 437004) and in mouse bone marrow samples using the Mouse 1β ELISA MAX™ Deluxe ELISA kit (Biolegend 432601) according to manufacturer's instructions. Absorbance at 450 nm was read using a plate reader (Varioskan LUX multimode reader, ThermoFisher Scientific) and Gen5 Data Analysis software.

**Immunohistochemistry**. Immunohistochemistry was performed on a Leica Bond using standard protocol F with increased primary antibody incubation time and without antigen retrieval. Antibodies used were: IL1β (1:500, abcam ab9722).

**Signalling arrays**. Assessment of signalling pathways in breast cancer cells following IL1β treatment was performed using Cignal 45 Pathway Reporter Arrays (Qiagen) as per manufacturer's instructions. Cells were either treated as control (media only) or with 10 ng/ml IL1β for 16 h. The luciferase assay was then carried out in Dual-Glo Luciferase assay system as per manufacturer's instructions. Data were analysed using the provided Data Analysis Template (Qiagen).

**In vivo experiments**. For subcutaneous experiments, 1x10⁶ MCF-7 cells were injected into each flank of 12 week old female NSG (NOD.Cg-Prkdcscid Il2rgtm1Wjl/SzJ, Charles River) mice. Mice were Supplemented with 0.18 mg oestrogen pellets (Innovative Research America) and euthanised when tumours reached a total tumour volume of 1.3 cm³. Tumours were digested using the Miltenyi tumour dissociation kit (130-095-929) as per manufacturer's instructions. For intra-femoral injections, $5 \times 10^5$ dual tdTomato and luciferase (pFULT, kindly provided by Dr Gambhir, Stanford University, CA[43]), expressing MCF-7 cells in 10 µl PBS were injected directly into the femurs of 8 week old female NOD/SCID mice using a Atraucan® single shot needle (B Braun Medical Ltd) (5 control (untreated) mice, 10 treated mice, both femurs injected). Mice were not supplemented with oestrogen. Mice were either treated with Vanticumab (15 mg kg⁻¹ weekly by intra-peritoneal injection) or control IP (sterile PBS) from 3 days prior to cell injections and for the duration of the experiment. Bioluminescence imaging was carried out using a Bruker In-Vivo Xtreme imaging system. All mice were culled between 6 and 8 weeks following cell injection. Femurs were collected in RPMI on ice, and washed in 70% ethanol followed by three washes in sterile PBS. Cells were flushed from femurs into cold RPMI using a 25 G needle. Bone ends were crushed with a scalpel and passed through a 100uM filter into RPMI. Cells were pelleted at 500 G for 7 min, and cells were counted based on tdTomato expression before plating as mammospheres. For spontaneous bone metastases experiments, tumour cells were pre-labelled with 25uM DiD (1,1'-dioctadecyl-', 3'-tetramethylindodicarbocyanine, 4-chlorobenzenesulfonate.Life Technologies, UK) for 15 min[44]. $5 \times 10^5$ GFP and luciferase labelled MDA-MB-231_BH cells were injected intravenously via the tail vein into 8 week old female BALB/c Nude mice ($n = 10$/group) (this cell line is bone seeking following tail vein injection[16]). Mice were monitored for a period of 30 days following cell injection and all culled on Day 31. Mice were treated with either an antibody to cleaved IL1β (1400.24.17, Invitrogen, UK; 2 mg kg⁻¹ subcutaneously three times weekly) or sterile PBS, or Vanticumab (15 mg kg⁻¹ intra-peritoneal injection weekly) or sterile PBS. Injections started prior to tumour injection (day −1 anti-IL1β antibody, day −3 Vanticumab) and continued throughout the experiment. Mice were imaged for tumour development by GFP/luciferase weekly using an IVIS imaging system, and hind limbs were extracted for individual ex vivo imaging on termination of the experiment. Right hind limbs were flash frozen in liquid N₂ for two-photon analysis of tumour cells within bone. Left hind limbs were fixed in 4% PFA for microcomputed tomography (µCT) analysis before decalcification in 1%PFA/0.5% EDTA and processing for histology. The presence of tumours in hind limbs was determined by visual examination of H&E stained slides by two independent scientists. µCT analysis assessed trabecular bone volume (BV) as a percentage of total tissue volume (TV) (BV/TV).

For the PDX experiment, BB3RC32 cells (ER+, PR+, Her2− PDX as described in[26]) were labelled with luciferase by electroporation and $1 \times 10^5$ cells injected into the left cardiac ventricle of 6 week female NSG mice (12 mice/group). Cells were not selected for luciferase expression prior to injection, due to the risk of damage to PDX cells. Mice were treated with Anakinra (1 mg kg⁻¹ subcutaneously) or sterile PBS daily until mice became moribund (58 days post tumour injection). Mice were monitored for tumour development and tumours were detected using methods detailed above. DiD cell two-photon analysis could not be carried out in PDX cells, due to these cells not taking up the DiD label, therefore the presence of disseminated tumour cells in bone was assessed by two-photon imaging of phycoerythrin-conjugated cytokeratin 19 (sc376126, Santa Cruz Biotechnology).

**Two-photon microscopy**. Tibiae were imaged using a multiphoton confocal microscope (LSM510 upright; Zeiss, Cambridge, UK). DiD labelled cells were visualised using a 633-nm visible laser, bone was detected using a 900-nm multiphoton Chameleon laser (Coherent, Santa Clara, CA). Images were reconstructed in LSM software version 4.2 (Zeiss) and numbers of cells per mm³ were counted using Volocity 3D Image Analysis software 6.01 (PerkinElmer, Cambridge, UK).

**DKK1 survival analysis**. Three publicly available Affymetrix datasets (MSK82 [GSE2603][45], EMC192 [GSE12276][46], EMC286 [GSE2034][47]) of primary breast tumours with information on the site of metastasis were integrated following RMA normalisation and the Combat[48] batch correction approach and duplicates removed. Similarly gene expression data for 2999 primary breast tumours from 17 published Affymetrix studies were integrated as described previously[49]. MAS5 detection calls[50] were used to define in which samples DKK1 was called 'present'. Associations with outcome were assessed by Kaplan Meier analysis using the *Survival* R package[51].

**Statistical methods**. Data are represented as mean ± SEM taken over a minimum of three independent experiments with three technical replicates per experiment, unless otherwise stated. For mammosphere experiments each independent experiment contained six technical replicates, and all replicates were included in statistical analyses. Unless stated otherwise, statistical significance was measured using parametric testing, assuming equal variance, with standard unpaired t-tests used to assess difference between test and control samples. Differences were considered statistically significant if the two-tailed probability value ($p$) was ≤0.05. All graphs represent mean ± SEM, *$p < 0.05$, **$p < 0.01$, ***$p < 0.001$, ****$p < 0.0001$.

**Reporting summary**. Further information on research design is available in the Nature Research Reporting Summary linked to this article.

## Data availability

Data and metadata on PDX models are available in PDX Finder (pdxfinder.org) and in the EurOPDX data portal (http://dataportal.europdx.eu). All other data that support the findings of this study are available from the corresponding authors upon reasonable request.

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

## Acknowledgements

This study was funded by Breast Cancer Now (Grant numbers: 2016MayPR734, MAN-Q1and MAN-Q2), NIHR Manchester Biomedical Research Centre (IS-BRC-1215-20007), EdiREX Horizon 2020 grant No.731105, MRC-MR/P000096/1 (PDO and CT) and Weston Park Hospital (PDO and DL). We would like to thank all patients who

donated tissue to this study, and the Manchester Cancer Research Centre Biobank for consenting patients and collecting tissue. We would like to thank Simon Mendez-Ferrer and Ya-Hsuan Ho (Wellcome - MRC Cambridge Stem Cell Institute, University of Cambridge) for helpful discussions.

## Author contributions

R.C., R.E., S.H., G.F. conceived the project; R.C., R.E., P.O., G.F., M.B., A.S., J.M. conceived and designed the analysis; R.E., D.A., A.S.G., K.S., J.M., C.H., B.S., D.L., C.T., J.S. collected the data; A.G., N.C., S.H. contributed data or analysis tools; R.E., D.A., A.S.G., K.S., J.M., D.L., C.T., P.O. performed the analysis; R.E., S.H., A.S., P.O., G.F., R.C. wrote the paper.

## Competing interests

A.G. is Chief Scientific Officer at Oncomed Pharmaceuticals. All other authors declare no competing interests.
