## [Peer Review File · Nature Communications]

Reviewers' comments:

Reviewer #1 (Remarks to the Author):

In the revised manuscript, the authors addressed all the concerns of this reviewer. The work is novel and timely and it further advances our understanding of breast cancer metastasis to bone.

Reviewer #2 (Remarks to the Author):

In the revised manuscript, the authors have addressed many of my comments. Specifically, they revised and toned down some of their statements (regarding cancer cell stemness, the importance of hematopoiesis, neutrophils etc.) to make them more consistent with the actual data. This is reflected also in the change of the manuscript title. In addition, the authors provided essential information on data acquisition and analysis that was missing before, and improved data presentation. As a result, the manuscript is much improved.

However, several important concerns still need to be addressed before publication:

1. The entire study is still based on in vitro experiments and experimental metastasis assays, which do not mimic the multistep process of spontaneous metastasis. This is particularly problematic as the authors are trying to address questions of tumor cell dormancy in the bone, which cannot be recapitulated when injecting large amounts of cancer cells systemically.
2. Moreover, all the experiments are performed with human cell lines/xenografts in immune deficient mice. The authors responded to my comment by explaining "carrying out experiments in immune competent mice models was beyond the scope of this current manuscript". In fact, performing experiments with a syngeneic breast cancer cell line in immune competent mice (e.g. 4T1) would not involve more work than performing the new PDX experiments which the authors performed in the revised study. This point is central since the authors conclude from their study that the use of adjuvant therapy targeted to the pathways identified would be beneficial in preventing bone metastasis. However, since they provide no insight on the role of the immune system, this conclusion is not backed up by the findings.
3. The authors provide no insight on the cellular source of IL-1b, and wrote in their rebuttal that they plan to address this in future work. However, "bone marrow" is very general. FACS sorting cells from BM of mice and analyzing the expression of IL-1b in different cell types is a straightforward experiment that can be included in the current study.

Reviewer #3 (Remarks to the Author):

Eyre et al. have taken on board the vast majority of the comments put forward by the reviewers previously. In transferring the manuscript it has also allowed them to include additional data and also discuss the findings and implications in more detail. The topic still remains of interest and is timely and the new mechanistic data will certainly likely add to the field. The addition of the additional cell lines and also the PDX model is a definite plus. The experiments showing the exogenous addition of IL1b to MCF7 and MDA231BH and the effects on mammosphere formation, along with validation of the phosphoCREB and phosphoNFkB signaling have also added to the manuscript. In general the manuscript has improved however, I have a few concerns that should still be addressed.

In my previous comment I enquired about the driver behind the 'activation' of the BM, and the secretion of IL1b from the BM, as well as the fact that in their cultures the BM have never seen cancer cells, nor the secreted factors from cancer cells. It is widely accepted that systemic changes in the BM occur in tumor bearing patients prior to tumor cell arrival, but this is something that has not been addressed. This also echoed by reviewer 3 and also other reviewer comments that the length of time in vitro required by the BM before it becomes 'stimulatory' is very long (>3 weeks

on plastic). Whilst there is in vivo data to support the story overall, there is a concern that this could be an artefact of the tissue culture situation, or just that it takes that long for a specific subpopulation of BM derived IL1b secreting cells to differentiate and take over the culture such that the BM elicits the observed effects.

I accept that obtaining bone marrow from tumor bearing patients may be technically challenging and not possible for these researchers at this moment in time. However, I believe that it is an important experiment that the authors conduct whereby they incubate the harvested 'normal' BM from healthy patients with CM collected from tumor cells (especially the ones they are using in their models) to mimic the in vivo setting of long-range signaling to the bone prior to colonization. This would reflect the situation that would occur in cancer patients and also occurs in their in vivo orthotopic models. This would essentially "prime" the BM similar to a tumor bearing patient and allow them to address my and other reviewers concerns. The downstream experiments using this 'primed BM' should show the same as the data they already present and this would greatly support their conclusions. In the very least, the authors should show that this priming leads to an increase in IL1b secretion.

This also links to one of their rebuttal comments where the authors themselves say that "Alternatively it may be that tumour cells somehow promote the production of IL1b by the BM". The above experiment would be very simple to undertake and directly answer this question. It may also be that different types of breast cancer induce IL1b secretion to a different degree. If this were the case, it would go some way to unpicking why ER+ and ER- breast cancer cells appear to colonize the bone with different efficiencies.

As previously requested, in order to validate the primary tumor DKK1 patient data and the hypothesis that DKK1 status of primary tumors cells is important in influencing bone colonization (as well as supporting the supporting vancticumab data), and rather than just discussing it, a simple experiment to show that DKK1 depletion in the cells (which they already have) and the effects on bone colonization should be included. This functional validation would support the inclusion of the tumor specific DKK1 role, since vancticumab will inhibit Wnt systemically and so the effects they see may not entirely be tumor cell DKK1 status dependent.

When assessing trabecular and total bone volume by micro-CT, have the authors used naïve mice as control? It does not appear to be the case. There is typically a large variation in TV and BV between mice especially those <10 weeks of age. Even age and litter matched mice exhibit high levels of inter-mouse variability (2-3%), and so this would be needed as a control in these experiments. With such small differences it is hard to be sure that this data is correctly powered as well, especially for example in Fig 5B where the difference between vehicle treated and IL1b NA treated is 8.5% vs 11% +/- what looks like 1-2%, and in Supp5E the difference is 7% BV/TV vs 9% BV/TV +/- 1%.

Further to this, it is not correct to assume that an "increase" in TV is as a result of dormancy in the absence of a naïve control to show where starting bone volume/trabecular volume lies. It could simply be that the BV/TV is simply not 'decreasing' by the 1-2 % shown in the graphs. A measure of loss of TV in treated compared to non-tumor bearing mice would then allow the assessment of blocking this tumor driven event.

The authors should show the calculated fold changes in the cytokine array data to Figure 3 since their choice of targets was queried by 3 reviewers and the addition of this data will help their case that whilst it was not the highest expressed, it was the largest change.

In the PORCNI and IL1b NA experiments, the inhibitors would still be present in the CM collected from the BM which was then applied to the cancer cells. Thus it is not clear how the authors can be sure that the effects are solely due to inhibiting the BM production of the Wnt or IL1b. Similarly, the pre-treatment of the cancer cells for 72 hours followed by inhibitor washout prior to adding BM CM does not rule out production by the cancer cells in response to the BM, unless they can categorically prove that this inhibition lasts for the duration of the BM CM exposure (although this would certainly not be the case for the IL1b NA study). Can the authors explain why they did not concurrently treat the cancer cells with BM derived CM, and PORCNI / IL1b NA? Some carefully

planned additional control experiments are warranted here to ensure that the conclusions they draw are sound.

It is not clear the relevance of the anoikis experiments are since the bone marrow is certainly not considered an attachment-free (anoikis inducing) environment. Unless the authors are arguing that metastatic dissemination through the blood is selecting for IL1R+ CSCs that can then colonize an IL1b rich BM environment, however this would then negate all of their intra-femoral bulk injection experiments where they have not selected for IL1R+ CSCs, although this may in part explain why the intra-femoral results with vantiactumab were inconclusive.

The authors need to take more care with their terminology. "Spongy bone" is the same as "trabecular bone". On P14L295-296 the authors state that Wnt inhibition increased trabecular bone but reduced spongy bone. This sentence directly contradicts itself and makes one wonder what exactly they are measuring in their analyses.

When discussing the DiD labelled tumor cell experiment in figure 5, the authors state that a similar number of tumor cells were present in the bone at 28d, but that vantiactumab drives dormancy. How is it possible that the number of tumor cells can be similar, yet in the treated mice the tumor cells can be held in a dormant state compared to untreated? As DiD labelled cells divide, the signal in halves which means it should dilute out and disappear as tumor cell number increases. The authors should clarify this issue.

In addition to this, it appears that no DiD labeling experiments were done with the PDX model, therefore it is not possible for the authors to conclude that the PDX tumor cells are still present in the bone at endpoint. It is equally likely that they could have apoptosed or been cleared, which is why there are no overt tumors present. Given the ER+ vs ER- state of the PDX and MDA231BH experiments, it is dangerous to assume that both are undergoing the same dormancy process without further experimental work.

Were the PDX cells transfected with luciferase selected or sorted to ensure 100% expression? If yes, please give details. If not, a mixed population could significantly alter bioluminescent signal readouts as different subpopulations grow out.

Minor comments

In response to a reviewer 3 comment about the dormant tumor cells and why only 1/3rd of patients relapse in the bone, an important fact for the authors to keep in mind is that there is a flip side to their argument that IL1b is upregulated. That is that an inhibitory factor may be lost that allows escape from dormancy. For example, there has been some very elegant work showing this is the case for Thrombospondin 1 in the perivascular niche.

It could be useful to add the T47D vantiactumab data sent to reviewers to supplementary. If the authors choose to publish the reviews, this will be available anyway.

It is very difficult to see the 1.5x fold change in the figure 4D. The scale should be changed to see this better as there appears to be a lot of heterogeneity so it is not possible to see if it complements the MCF-7 data. In addition to this, have the authors profiled any of the other lines they use in the manuscript?

Does inhibiting both CREB and NFkB give increased benefit of either alone?

The figure 4G western blot appears as though the left-most lane (NFkB p65 Cont) has had a serious issue and not run/transferred/developed correctly. Please provide a better quality blot.

Consider moving the data from 2E to supplementary as it does not really support the story

P9L188 – the data presented in the manuscript does not mechanistically support the conclusion that "high-Wnt signaling tumors are those which are more able to colonize the bone environment".

Without the above suggested DKK1 KO in vivo study this conclusion needs to be moderated back.

P10L206 and L203 – PORNi should probably read PORCNI

P12L257 – “Ascitic” fluids

In the discussion, the authors could link their discussion of the 30% bone metastases back to their data on DKK1 status. i.e. DKK1 at the primary tumor is likely low in the CSC-like population which then facilitates the metastatic colonization in the bone at a later point.

In the methods the authors refer to the MDA213BH and MDS231IV – Are these the same? Be consistent with your naming.

Figure 2D – the contrast of the IVIS photographs is a little difficult to see the mice. The luminescent signal appears OK.

Figure 4D – no axis on graph

Reviewer #4 (Remarks to the Author):

In current manuscript, Eyre and colleagues showed bone marrow culture derived IL1beta induce Wnt autocrine signaling in breast cancer cells thereby increase breast cancer bone colonization. The findings are potentially interesting, however following concerns needs to be addressed:

It is impressive that the effect of bone marrow condition medium on mammosphere formation is preserved across many early breast cancer patient samples. However, it is not clear why use early breast cancer samples and how these samples relate to bone metastasis? It is also clear how 4 out of 17 samples were chosen for mammosphere self-renewal assay.

Both mammosphere formation assay and Aldeflour assay suppose to measure breast cancer stem cell property. It is not clear why bone marrow CM has opposite result on MCF7 in these assays.

Seems the effect of Vantictumab is limited to in vitro and ex vivo experiments but not in vivo experiments, which does not support the main hypothesis. Furthermore, it is also intriguing that Wnt inhibition increased trabecular bone volume but reduced cortical and spongy bone. It can be helpful to clarify what it means and if it is tumour dependent.

Seem the in vivo effects of IL1beta inhibition and Wnt inhibition are different. IL1beta inhibition seems to affect bone homing while Wnt inhibition does not. This seems contradict with the authors' model that these two pathways work in a linear fashion. Furthermore, the fact that Wnt inhibition leads to dormancy in vivo needs to be proved directly as the effect can be also explained by reduced proliferation.

It is critical to show if the proposed model is relevant to the disease in patients. The authors showed that DKK level is associated with bone metastasis free survival. However, it can be important to show if there is any connection between IL1beta and DKK1/Wnt pathway in patient dataset and if IL1beta level is also associated with bone metastasis free survival in order to support the proposed model?

Response to reviewers' comments

We thank all the referees for their efforts reviewing the revision of our manuscript, and for making very constructive criticisms which we have addressed in this second revision as detailed below. We believe this has led to a better manuscript and has clarified the data in many respects.

We have obtained, analysed and present new data in the paper in response to these comments, including:

- 1) Immune staining and ELISA analysis of IL1 β in the bone marrow of immune-competent and immune-deficient mice demonstrating it to be a commonly expressed cytokine in several mouse models.
- 2) Analysis of published single cell data from bone marrow showing IL1 β expression in multifarious immune and non-immune cell types.
- 3) Demonstration that conditioned medium from breast cancer cells induces IL1 β secretion from bone marrow *in vitro*.
- 4) Measuring DKK1 expression in bone homing cells and demonstrating it to be lower in parental MDA-MB-231 cells, confirming its contribution to the homing phenotype.
- 5) Analysis of cortical bone volume in control and Vantictumab treated mice to show a loss of bone volume in mice following anti-Wnt treatment.
- 6) 2 photon imaging of disseminated cells in bone marrow *in vivo* following Anakinra treatment, demonstrating that they are present but prevented from forming metastatic colonies.
- 7) Wnt ligand profiling of the MDA-MB-231_BH bone homing cell line in response to treatment with IL1 β .
- 8) Mammosphere colony forming data showing the synergistic effect of drug combinations targeting both NF κ B and CREB signalling pathways.

We believe these additions address important comments raised by the reviewers of the previous version, and therefore substantially strengthen the paper and the conclusions of the study.

Reviewer #1 (Remarks to the Author):

In the revised manuscript, the authors addressed all the concerns of this reviewer. The work is novel and timely and it further advances our understanding of breast cancer metastasis to bone.

We thank the reviewer for their support for publication of our findings.

Reviewer #2 (Remarks to the Author):

In the revised manuscript, the authors have addressed many of my comments. Specifically, they revised and toned down some of their statements (regarding cancer cell stemness, the importance of hematopoiesis, neutrophils etc.) to make them more consistent with the actual data. This is reflected also in the change of the manuscript title. In addition, the authors provided essential information on data acquisition and analysis that was missing before, and improved data presentation. As a result, the manuscript is much improved.

We would like to thank the reviewer once again for their helpful reviews, and we agree that the manuscript is much improved thanks to their previous suggestions. The reviewer continues to raise important points in this second round of reviews following transfer of the manuscript to Nature Communications. These comments are responded to individually below, and clarified in the manuscript text.

However, several important concerns still need to be addressed before publication:

1. The entire study is still based on *in vitro* experiments and experimental metastasis assays, which do not mimic the multistep process of spontaneous metastasis. This is particularly problematic as the authors are trying to address questions of tumor cell dormancy in the bone, which cannot be recapitulated when injecting large amounts of cancer cells systemically.

We agree with the reviewer that a limitation of this study is its reliance on *in vitro* experiments and experimental metastasis assays *in vivo*. Studying metastasis *in vivo* is difficult, and no single model has been identified which accurately recapitulates human breast cancer metastasis.

Given the limitations of the models currently available, we have tried to select the most appropriate models available for our individual experiments. None of our experiments were designed to address the multiple steps of spontaneous metastasis, but were designed to test colony formation following the arrival of breast cancer cells in the bone microenvironment. The study of dormancy rather than colonisation is even more challenging (both *in vitro* and *in vivo*), therefore in all our experiments we have elected to address colony-forming activity (rather than dormancy) of breast cancer cells, with sphere forming assays from single cells *in vitro*, and *in vivo*. To make this clearer throughout the manuscript, we have removed specific mentions of the word dormancy from our Materials and Methods and Results sections, and instead only refer to colony forming activity. Further, where we had previously interpreted our multi-photon data as Vantictumab promoting tumour cell dormancy *in vivo* (Figure 5E), we have changed the conclusion surrounding this experiment to read “28 days following injection similar numbers of tumour cells were present in the proximal tibiae of control or Vantictumab treated mice (Figure 5E), demonstrating that although inhibiting Wnt signalling prevents colonisation of cells to form bone metastasis, it does not prevent tumour cells homing to bone” which we think better reflects the data we have shown. We would like to thank the reviewer for this comment, as this has improved clarity of our results throughout the manuscript, and allowed us to make clearer the specific focus on colonisation rather than dormancy.

A salient point not previously discussed is that the ER+ PDX model we used in our first paper revision (experiment using intra-cardiac injection of PDX cells is shown in Figure 5F) spontaneously metastasises to bone following orthotopic injection. This was one of the major reasons for selecting this model (images below showing mammary tumours following orthotopic injection of luciferase positive PDX cells into the 4th and 9th mammary glands of 12-week-old female NSG mice (A), and subsequent metastases to mouse tibiae (B)).

However, spontaneous metastasis of PDX cells to bone following orthotopic injection requires both a long experimental time period (approximately 4-6 months) and larger numbers of mice (due to a lower metastasis efficiency). We therefore elected to introduce these PDX cells via intra-cardiac injection for the experiment included in this manuscript for reasons of feasibility. We have now added a sentence stating that this PDX model spontaneously metastasises to the Methods section on Page 31 (highlighted in manuscript text).

To make our model selection, and the caveats of existing metastasis models, clearer, we have added a section to our discussion focussing on the limitations of current metastatic models and the need to test our hypotheses in spontaneous *in vivo* metastatic models to assess the effect of IL1 β -Wnt inhibition on the entire metastatic cascade. This can be found in the manuscript on Page 19/20 (highlighted in manuscript text).

2. Moreover, all the experiments are performed with human cell lines/xenografts in immune deficient mice. The authors responded to my comment by explaining “carrying out experiments in immune competent mice models was beyond the scope of this current manuscript”. In fact, performing experiments with a syngeneic breast cancer cell line in immune competent mice (e.g. 4T1) would not involve more work than performing the new PDX experiments which the authors performed in the revised study. This point is central since the authors conclude from their study that the use of adjuvant therapy targeted to the pathways identified would be beneficial in preventing bone metastasis. However, since they provide no insight on the role of the immune system, this conclusion is not backed up by the findings.

Following the first round of reviews, Reviewer 2 suggested carrying out experiments in an immune-competent mouse model, whereas Reviewer 3 suggested the use of a PDX model.

Following discussion amongst the authors, we decided to undertake additional *in vivo* experiments for this manuscript in a spontaneously metastatic ER+ PDX model, as we felt that this was the best fit with the data already included. This decision was made for two reasons: Firstly, by using a PDX model we were able to investigate the effect of inhibiting IL1 β signalling in an ER+ *in vivo* model, which would not have been possible in an immune-competent setting. Secondly, we felt that including results from the human PDX model provided a natural extension to our *in vitro* work, which has been carried out using patient-derived breast cancer and bone marrow samples.

We agree with the reviewer that understanding the role of the immune system in this signalling pathway is essential before we can draw firm conclusions on the use of these inhibitors as adjuvant therapy in breast cancer. We apologise for our previous use of the statement “carrying out experiments in immune competent mice models was beyond the scope of this current manuscript”, as this was too ambiguous, and did not provide sufficient insight into the results we already have on this subject. We have already carried out experiments assessing the effect of IL1 β inhibition with Anakinra in both 4T1 and E0771 immune competent models. In this work, we found reduced spontaneous bone metastasis and reduced outgrowth of tumour cells disseminated in bone following daily administration of Anakinra. A summary of this data is available in a published abstract (Tulotta, et al. Breast Cancer Research and Treatment (2018) 167:309-45 P9.4), and graphical representation of the E0771 data is shown below.

[redacted]

We have elected not to include this *in vivo* data in our current manuscript, as this forms part of our larger follow up studies into the role of the immune system in promoting IL1 β -Wnt signalling and metastases. The role of the immune system in metastasis is a vast topic, and providing a comprehensive answer on the role of IL1 β -Wnt signalling within this is a long term study, particularly given the rapidly increasing number of studies demonstrating roles for IL1 β in cancer immunology (including the new paper by Kaplanov et al (PNAS, 2019; 116 (4) 1361-1369) which shows that inhibiting IL1 β in mouse models enhances tumour cell

immunity by synergising with anti-PD-1). Our current manuscript establishes a role for IL1 β in metastasis independent of the immune system (such as the activation of CSCs within the bone metastatic environment, and the downstream signalling pathway in cancer cells activated by IL1 β), and we believe it makes sense to publish this independently of our work on the role of IL1 β in the immune system.

To further demonstrate this immune cell-independent role for IL1 β , we have examined levels of IL1 β in immune competent and immune deficient mice bones by ELISA and by IHC. To assess IL1 β in bone marrow by ELISA, bone marrow was flushed from the tibiae and femurs of 10-14 week old female immune competent (BALB/c) and immune compromised (BALB/c nude and NOD SCID Gamma (NSG)) mice. Protein was extracted from the marrow of n=6-8 mice per group and run on an ELISA plate for IL-1 β . IL1 β was easily detected in NSG and nude immune deficient mice and there was a small but significantly increased expression in BALB/c immune competent compared to NSG immune deficient mice (see graph below, data are mean +/- SD of IL-1 β concentrations normalised to ug of total protein. p=0.01).

These ELISA results are supported by IHC staining of femurs from immune competent BALB/c mice, which have stronger IL1 β staining than femurs from immune deficient NSG mice (see figure below). The ELISA and IHC data have now been added to the manuscript in Supp Figures 3D and 3E and are discussed in the manuscript on Page 11/12 (highlighted in manuscript text). Methods for these experiments have been added on Page 28 (highlighted in manuscript text).

We have therefore expanded the discussion section on the immune response (Page 20, highlighted in manuscript text). Within this, we have clarified that our conclusions have been derived from work in immune deficient mice, and have added the caveat to our conclusions at the end of the manuscript that our results suggest inhibiting IL1 β -Wnt signalling could be useful in breast cancer, but results would need to be confirmed in immune-competent animals prior to this being considered adjuvant therapy in patients (highlighted in manuscript text on Page 21).

3. The authors provide no insight on the cellular source of IL-1b, and wrote in their rebuttal that they plan to address this in future work. However, “bone marrow” is very general. FACS sorting cells from BM of mice and analyzing the expression of IL-1b in different cell types is a straightforward experiment that can be included in the current study.

We agree with the reviewer that determining the cellular source of IL1 β from bone marrow promoting metastatic colonisation is important for our onward work, however this is not the straightforward experiment suggested. There are many published studies showing the cells within bone marrow which express IL1 β , and these studies demonstrate that there are multiple IL1 β expressing populations within bone marrow. These studies (amongst others) have identified the following bone marrow cells expressing IL1 β : macrophages (Wong et al 2014, *Cancer Biology & Therapy*. 15 (10) 1395-1403. Frisch et al 2019, *JCI Insight*; 4(10)); monocytes, macrophages and dendritic cells (de Mooij, et al, *Blood*. 2017;129(24):3155-3164); monocytes and neutrophils (Schreiber et al 2012, *J Am Soc Nephrol* 23: 470–482); bone marrow-derived haematopoietic cells (Cho et al 2012, *PLoS Pathog* 8(11)), neutrophils (Karmakar et al 2015. *J Immunol*; 194:1763-1775).

Although the studies cited above have mainly identified IL1 β as being produced by immune cells, we have previously demonstrated production of IL-1 β by osteoblasts and haematopoietic cells (Tulotta et al, *Clin Can Res*. 2019). In addition, a very recently published paper on the cellular taxonomy of bone marrow stroma (Baryawno 2019, *Cell* 177, 1915–1932) which excluded immune cells by sorting cells negative for immune lineage markers (CD45/CD3/B220/CD19/Gr-1/CD11b), confirmed IL1 β to be expressed by non-immune cells within bone marrow stroma. In this study, single cell RNA-seq was performed on bone marrow stroma, and 18 distinct cell subsets were defined. We analysed IL1 β expression using this dataset and it is expressed by 14 of the 17 cell subsets (not present in subsets 11, 12 and 14). Therefore, in addition to immune cells, IL1 β is also expressed in bone marrow stroma by MSCs (cluster 1), MSC-descendent osteolineage cells (clusters 7 and 8), chondrocytes (clusters 2, 10, 13, and 17), fibroblasts (clusters 3, 5, 9, 15, and 16), bone marrow derived endothelial cells (clusters 0, 6, and 11), chondrocytes and osteoblasts (clusters 4 and 8) (see tSNE plot below of IL1 β expression. X axis; cluster number).

This published data demonstrates that the cellular source of IL1 β in bone marrow has already been extensively investigated, and IL1 β is expressed by multiple cell populations. We have added a statement explaining the complexity of IL1 β expression in bone marrow on Page 18 (highlighted in manuscript text), and the importance of understanding this in future studies.

Reviewer #3 (Remarks to the Author):

Eyre et al. have taken on board the vast majority of the comments put forward by the reviewers previously. In transferring the manuscript it has also allowed them to include additional data and also discuss the findings and implications in more detail. The topic still remains of interest and is timely and the new mechanistic data will certainly likely add to the field. The addition of the additional cell lines and also the PDX model is a definite plus. The experiments showing the exogenous addition of IL1b to MCF7 and MDA231BH and the effects on mammosphere formation, along with validation of the phosphoCREB and phosphoNFkB signaling have also added to the manuscript. In general the manuscript has improved however, I have a few concerns that should still be addressed.

We would like to thank this reviewer for another constructive and fair review following our transfer of this manuscript to Nature Communications. In the first round of reviews, we modified the manuscript significantly following comments from this reviewer, and feel it is much improved as a result. Similarly, in this second round of reviews, the reviewer has raised additional points for further clarification, and we have modified the text and included further data to the figures to incorporate these. Individual comments are discussed below.

In my previous comment I enquired about the driver behind the 'activation' of the BM, and the secretion of IL1b from the BM, as well as the fact that in their cultures the BM have never seen cancer cells, nor the secreted factors from cancer cells. It is widely accepted that systemic changes in the BM occur in tumor bearing patients prior to tumor cell arrival, but this is something that has not been addressed. This also echoed by reviewer 3 and also other reviewer comments that the length of time in vitro required by the BM before it becomes 'stimulatory' is very long (>3 weeks on plastic). Whilst there is in vivo data to support the story overall, there is a concern that this could be an artefact of the tissue culture situation, or just that it takes that long for a specific subpopulation of BM derived IL1b secreting cells to differentiate and take over the culture such that the BM elicits the observed effects.

I accept that obtaining bone marrow from tumor bearing patients may be technically challenging and not possible for these researchers at this moment in time. However, I believe that it is an important experiment that the authors conduct whereby they incubate the harvested 'normal' BM from healthy patients with CM collected from tumor cells (especially the ones they are using in their models) to mimic the in vivo setting of long-range signaling to the bone prior to colonization. This would reflect the situation that would occur in cancer patients and also occurs in their in vivo orthotopic models. This would essentially "prime" the BM similar to a tumor bearing patient and allow them to address my and other reviewers concerns. The downstream experiments using this 'primed BM' should show the same as the data they already present and this would greatly support their conclusions. In the very least, the authors should show that this priming leads to an increase in IL1b secretion.

Thanks to the reviewer for this comment. We have previously published that culturing human breast cancer cells in pieces of whole human bone for 48 hours results in increased secretion of IL1 β (see Figure 5a, Tulotta et al, Clinical Cancer Research 2019; 25 (9) 2769-

2782), suggesting that direct tumour cell-bone interactions promote production of IL1 β that can drive expansion of the niche and stimulate the formation of overt metastases. However we had not previously investigated long range signals from primary tumour cells to bone marrow. We have therefore performed the *in vitro* experiment suggested by the reviewer, treating bone marrow cultures with tumour cell conditioned media to model the pre-metastatic niche. In this experiment we took 3 flasks of a bone marrow sample growing in culture, then treated these bone marrow samples with conditioned media from MCF7 and MDA-MB-231_BH cells, and compared IL1 β levels in these cultures by ELISA compared to in bone marrow treated with control media at 24 hours, 72 hours, and 7 days after treatment (graph below).

In the untreated bone marrow sample (black line), IL1 β levels remained constant over the 7 day culture period, as would be expected. In the bone marrow samples treated with tumour cell conditioned media, a significant increase in IL1 β levels was seen. Interestingly, this effect was observed at different time points for the 2 different cell lines. When bone marrow was treated with MCF7 cell conditioned media the increase in IL1 β was seen after 72 hours (red line; p=0.0004), whereas in bone marrow treated with MDA-MB-231_BH conditioned media the increase in IL1 β was seen at the earlier time point of 24 hours (orange line, p=0.0306).

These data add weight to the hypothesis that the primary tumour secretes factors which activate the bone marrow to produce IL1 β , and we have added these to the manuscript in Supplementary Figure 6, and to the discussion on Page 20 (highlighted in manuscript text).

This also links to one of their rebuttal comments where the authors themselves say that “Alternatively it may be that tumour cells somehow promote the production of IL1b by the BM”. The above experiment would be very simple to undertake and directly answer this question. It may also be that different types of breast cancer induce IL1b secretion to a different degree. If this were the case, it would go some way to unpicking why ER+ and ER- breast cancer cells appear to colonize the bone with different efficiencies.

As shown in the data above, treating bone marrow in culture with tumour cell conditioned media induces IL1 β secretion. In the above experiment, we showed a greater induction of IL1 β secretion with conditioned media from the ER+ cell line compared to the ER- cell line, which may in part explain the difference between ER+ and ER- cells in efficiency of

colonising the bone marrow. Alternatively, this could also be explained by the different response of ER+ and ER- cells to IL1 β in terms of Wnt ligand production, which is higher in ER+ cell lines and patient samples (Figure 4C and 4D, additional data with an ER- cell line has been added to Figure 4C and is discussed in response to a comment below). We have added this to the Discussion on Page 17.

As previously requested, in order to validate the primary tumor DKK1 patient data and the hypothesis that DKK1 status of primary tumors cells is important in influencing bone colonization (as well as supporting the supporting vancticumab data), and rather than just discussing it, a simple experiment to show that DKK1 depletion in the cells (which they already have) and the effects on bone colonization should be included. This functional validation would support the inclusion of the tumor specific DKK1 role, since vancticumab will inhibit Wnt systemically and so the effects they see may not entirely be tumor cell DKK1 status dependent.

The role of DKK1 in this system was raised by the reviewer during the review at Nature, and following the reviewer's comments we removed data and toned down our hypotheses involving DKK1. In doing this, we have aimed to base our conclusions around the more general "Wnt signalling pathway" rather than the role of DKK1 specifically, with the specific role of DKK1 forming part of our ongoing investigations.

We would like to apologise to the reviewer for the confusion in the manuscript regarding our DKK1 knockdown cells. These cells are siRNA transient knockdown cells, not a stable knockdown. We have discussed if these siRNA transient knockdown cells would be suitable for injection *in vivo* as the reviewer suggests (with our hypothesis being that if we inject 231 cells with DKK1 knocked down, we would see an increase in bone colonisation compared to injection of parental 231 cells), however, given that our siRNA knockdown is not a complete knockout, and the effects of siRNA are not long-lasting, we are not confident that injecting these cells would give a large enough reduction in DKK1 levels to give a quantifiable reduction in metastatic colonisation over the 30 day experimental period. Based on this, we do not feel it would be scientifically justified to inject the siRNA DKK1 transient knockdown cells *in vivo*. We have added that the siRNA knockdown is for 72 hours to the Figure legend for Supp Figure 2 to clarify the transient nature of this knockdown.

To address the reviewer's comment, we have rearranged the manuscript to bring all our DKK1 data together (the existing clinical data is now presented with the DKK1 siRNA knockdown and mammosphere formation which was previously in Supplementary data). We have also added qRT-PCR for DKK1 mRNA from MDA-MB-231 (parental) and MDA-MB-231_BH (bone homing cell line). Our hypotheses surrounding DKK1 is that cells with lower levels of DKK1 (and thus increased Wnt signalling) will be more bone metastatic, and as predicted, MDA-MB-231 cells have significantly ($p=0.0031$) higher expression of DKK1.

All DKK1 data is now shown in Figure 2F-H. This has allowed us to add the overall conclusion: “These data add to our previous data indicating that Wnt signalling promotes the ability of CSCs to form colonies in the bone environment *in vitro*, and specifically suggest a role for DKK1 in promoting this, which warrants further investigation in *in vivo* studies.” This is highlighted in the manuscript text on Page 10. We have removed the conclusion “high-Wnt signalling tumours are those which are more able to colonize the bone environment”, as we agree with the reviewer that this comment does not directly reflect the results we have presented.

We agree with the reviewer’s comment that as vantiactumab inhibits Wnt systemically (because of the high degree of evolutionary conservation in this pathway, vantiactumab binds to both human and mouse Fzd receptors), we cannot assume that Wnt signalling *in vivo* is cancer cell autonomous. We have therefore added “*in vitro*” to our mechanistic conclusion in the discussion on Page 18 (highlighted in manuscript text), and have added a further explanation stating that “genetic knockdowns will also allow us to confirm *in vivo* our *in vitro* findings that the Wnt signalling driving cancer cell colonisation is cancer cell autonomous” on Page 19 (highlighted in manuscript text).

When assessing trabecular and total bone volume by micro-CT, have the authors used naïve mice as control? It does not appear to be the case. There is typically a large variation in TV and BV between mice especially those <10 weeks of age. Even age and litter matched mice exhibit high levels of inter-mouse variability (2-3%), and so this would be needed as a control in these experiments. With such small differences it is hard to be sure that this data is correctly powered as well, especially for example in Fig 5B where the difference between vehicle treated and IL1b NA treated is 8.5% vs 11% +/- what looks like 1-2%, and in Supp5E the difference is 7% BV/TV vs 9% BV/TV +/- 1%.

It is worth noting that the mice used in this experiment were 8 weeks old when tumour cells were injected, and therefore these mice were at least 12 weeks old when the bones were analysed. By 12 weeks, the mouse skeleton is mature and we observe much lower variability in BV/TV than in younger animals.

The control mice used in the *in vivo* experiments were all untreated mice which had tumour cells injected. We have made this clearer by adding this to the Figure legend for Figure 5 (highlighted in manuscript text) and to the manuscript text on Page 15 (highlighted in manuscript text).

Following the reviewers comment, we have additionally assessed BV/TV in a group of 12 week old female (i.e. age and sex matched, n=10) naïve mice using the same micro-CT machine used in the experiments shown in the manuscript. Results of this are shown in the graph below (white bar = tumour naïve mice. Purple bar = untreated mice from the IL1 β NA experiment. Red bar = untreated mice from the Vantictumab experiment. Data presented as mean \pm SEM). When BV/TV is compared between tumour naïve and tumour bearing untreated mice, although this looks lower in the tumour bearing mice, this was not significantly different (tumour naïve vs. untreated from IL1 β NA experiment p=0.1472, tumour naïve vs. untreated from Vantictumab experiment p=0.0561).

Our reason for using mice bearing tumour cells rather than naïve mice for our control is explained in the response to the comment below.

Further to this, it is not correct to assume that an “increase” in TV is as a result of dormancy in the absence of a naïve control to show where starting bone volume/trabecular volume lies. It could simply be that the BV/TV is simply not ‘decreasing’ by the 1-2 % shown in the graphs. A measure of loss of TV in treated compared to non-tumor bearing mice would then allow the assessment of blocking this tumor driven event.

Apologies to the reviewer, this is not what we were trying to suggest. Our reason for assessing BV/TV in (bone volume/total tissue volume) in mice which were untreated but with tumour cells injected vs. treated mice was to assess the effect of these treatments on bone, not to assess the effect of tumour cells on BV/TV.

The effect of the treatments on bone containing tumour cells is important, as metastatic breast cancer patients often experience bone loss as part of their disease, therefore any treatment which is bone building (in the presence of tumour cells) would be seen as an attractive option for these patients. Our results show that anti-IL1 β treatment increased bone volume compared to untreated bones containing tumour cells. We have rewritten this in the text on Page 15 to better explain the purpose of this experiment.

The authors should show the calculated fold changes in the cytokine array data to Figure 3 since their choice of targets was queried by 3 reviewers and the addition of this data will help their case that whilst it was not the highest expressed, it was the largest change.

The calculated fold changes for the top 10 cytokines which increased in expression from non-stimulatory to stimulatory CM are now presented as a table in Supplementary Figure 3C, and referred to in the manuscript text on Page 11 (highlighted in manuscript text).

In the PORCNi and IL1b NA experiments, the inhibitors would still be present in the CM collected from the BM which was then applied to the cancer cells. Thus it is not clear how the authors can be sure that the effects are solely due to inhibiting the BM production of the Wnt or IL1b. Similarly, the pre-treatment of the cancer cells for 72 hours followed by inhibitor washout prior to adding BM CM does not rule out production by the cancer cells in response to the BM, unless they can categorically prove this inhibition lasts for the duration of the BM CM exposure (although this would certainly not be the case for the IL1b NA study). Can the authors explain why they did not concurrently treat the cancer cells with BM derived CM, and PORCNi / IL1b NA? Some carefully planned additional control experiments are warranted here to ensure that the conclusions they draw are sound.

We would like to apologise for the confusion surrounding the methods for this experiment, which we have not stated clearly enough in the manuscript. For clarity, experiments using the PORCNi and IL1 β NA were set up in the following ways:

1. To test the effect of preventing Wnt/IL1 β secretion from bone marrow: Bone marrow growing in culture was treated with PORCNi/IL1 β NA. Conditioned media was taken from this bone marrow, and cancer cells in culture were then treated with this CM prior to assessment of CSC colony formation in mammosphere culture. CM from the same bone marrow but without PORCNi/IL1 β NA was used as a control for this experiment to show that CM without the addition of inhibitors promoted mammosphere formation (as we had shown previously).
2. To test the effect of preventing Wnt/IL1 β secretion from cancer cells: Cancer cells in culture were treated with PORCNi/IL1 β NA. These cells were then treated with bone marrow CM, prior to mammosphere culture. Breast cancer cells which had not been treated with inhibitors were used as a control for this experiment, to show that these responded to CM with increased mammosphere formation (as we had shown previously).

We have added extra details to the Figure legend (Figure 5) and have modified the manuscript text on Pages 11 and 12 (highlighted in manuscript text) to make these methods clearer.

To respond to the reviewers specific points on these experiments:

1. *“it is not clear how the authors can be sure that the effects are solely due to inhibiting the BM production of the Wnt or IL1b”*

In the case of the experiment where we have treated bone marrow with IL1 β NA and shown that this prevents response of breast cancer cells to bone marrow CM, we agree with the reviewer that as the IL1 β NA was still present in CM when the cancer cells were treated, it is possible that some of the effect of inhibiting IL1 β on mammosphere formation may have been due to inhibiting IL1 β in the cancer cells. However, our control for this experiment,

which was treating cancer cells with IL1 β NA prior to addition to bone marrow conditioned media (shown in the 4th bar in the graph presented in Figure 3H) showed that no reduction was seen in mammosphere formation compared to treatment with CM alone. Therefore we think it is unlikely that the effect seen when treating the BM with IL1 β NA was due to effect on cancer cells, as we would have also seen an effect in this group. To make this clearer, we have added “ns” to this bar in Figure 3H, and have modified the text surrounding this experiment on Pages 10-12.

2: *“the pre-treatment of the cancer cells for 72 hours followed by inhibitor washout prior to adding BM CM does not rule out production by the cancer cells in response to the BM”*

As the reviewer states, in this experiment we treated cancer cells with IL1 β NA prior to the addition of bone marrow CM. We had already shown that concurrent treatment of MCF-7 cells with bone marrow CM and IL1 β NA resulted in a reduction of mammosphere formation compared to treatment with CM alone (shown in Figure 2D), but this gave us no insight into whether the IL1 β being neutralised was present in CM or being produced in cancer cells in response to CM. Therefore we felt this consecutive treatment schedule was a way to demonstrate this. We have rewritten the text around this experiment to make this point clearer. We agree with the reviewer that based on these experiments we cannot entirely rule out the production of IL1 β by cancer cells (and conversely, Wnt ligands by BM), and to address this we have changed the sentence “confirming that Wnt ligands driving Wnt signalling and CSC colony formation are produced by cancer cells” to “indicating” on Page 11, and “this demonstrates that the IL1 β driving Wnt-dependent colony formation in breast CSCs *in vitro* is bone marrow derived, and not tumour cell-derived as suggested previously” to “indicates” on Pages 12 of the manuscript (highlighted in manuscript text).

It is not clear the relevance of the anoikis experiments are since the bone marrow is certainly not considered an attachment-free (anoikis inducing) environment. Unless the authors are arguing that metastatic dissemination through the blood is selecting for IL1R+ CSCs that can then colonize an IL1b rich BM environment, however this would then negate all of their intra-femoral bulk injection experiments where they have not selected for IL1R+ CSCs, although this may in part explain why the intra-femoral results with vantiactumab were inconclusive.

We apologise for the lack of clarity on this point. The reviewer is correct, we are hypothesising that metastatic dissemination is selecting for IL1R+ CSCs that colonise the IL1 β rich bone marrow. We have made this point clearer in the manuscript adding this point to Page 12 (highlighted in manuscript text). Thanks to the reviewer for highlighting this point to us.

With regards to the intra-femoral injection, as the reviewer suggests, we suspect that injecting this large population of bulk tumour cells may be why we did not see a reduction in tumour growth. We had already included a discussion of this on Page 19, but we have now modified this to make it clearer (highlighted in text on Page 19).

The authors need to take more care with their terminology. “Spongy bone” is the same as “trabecular bone”. On P14L295-296 the authors state that Wnt inhibition increased

trabecular bone but reduced spongy bone. This sentence directly contradicts itself and makes one wonder what exactly they are measuring in their analyses.

We would like to apologise to the reviewer for our misuse of terminology. In these analyses we have measured trabecular bone volume (BV) as a percentage of total tissue volume (TV). This was stated in the figure legends previously, but we have added this to the Materials and Methods section on Page 30 to improve clarity. We have removed the use of the term spongy bone from the manuscript.

In the sentence highlighted by the reviewer, we were trying to communicate that although BV/TV volume is higher in the Vantictumab treated mice, this is not due to a bone building effect, and should not be interpreted as such. BV/TV is higher because there are less bone metastases (and therefore less lytic lesions and bone destruction) in these animals. Vantictumab reduced cortical bone compared to control animals (mean %BV 0.321 in control animals vs 0.195 in Vantictumab treated animals, $p < 0.0001$, $n = 7$. Analyses undertaken in areas without tumours), and we have added this data to Supplementary Figure 5F (and referenced this to the manuscript text on Page 15) to further highlight this point.

When discussing the DiD labelled tumor cell experiment in figure 5, the authors state that a similar number of tumor cells were present in the bone at 28d, but that vantictumab drives dormancy. How is it possible that the number of tumor cells can be similar, yet in the treated mice the tumor cells can be held in a dormant state compared to untreated? As DiD labelled cells divide, the signal in halves which means it should dilute out and disappear as tumor cell number increases. The authors should clarify this issue.

We apologise to the reviewer that our explanation of this result was not clear. The vast majority of tumour cells that arrive in bone will remain dormant and will retain the DiD dye, because only a very small minority of cells produce metastatic outgrowth. DiD positive cells will therefore be present in bones of mice both with and without tumours. The conclusion we were trying to draw from our result is that treatment with Vantictumab did not prevent tumour cells from homing to bone (as a similar number of DiD labelled cells were present in the bone at 28 days), despite less overt metastases forming in these mice. We have rewritten the text surrounding this experiment in Page 16 (highlighted in manuscript text), and have removed speculation around dormancy to instead describe the result as Vantictumab preventing colonisation, but not tumour cell homing to bone.

In addition to this, it appears that no DiD labeling experiments were done with the PDX model, therefore it is not possible for the authors to conclude that the PDX tumors cells are still present in the bone at endpoint. It is equally likely that they could have apoptosed or been cleared, which is why there are no overt tumors present. Given the ER+ vs ER- state of the PDX and MDA231BH experiments, it is dangerous to assume that both are undergoing the same dormancy process without further experimental work.

We did attempt to DiD label the PDX cells prior to injection, but unfortunately these cells did not take up the DiD dye. This is why these analyses are not present for this experiment.

Following the reviewers comment, we have performed two-photon imaging to assess the presence of single disseminated tumour cells in bones of mice from the PDX experiments which did not develop overt tumours. An example of this immunohistochemical staining is shown below, where disseminated tumour cells were detected using phycoerythrin conjugated cytokeratin 19 (sc376126) (red) and bone is shown in blue.

Red cancer cells could be detected in the bones of Anakinra treated mice, showing that, as with the cell line experiments, tumour cells reached the bones, but did not grow into overt metastases. We have added these data to the manuscript in Supplementary Figure 5H, and this is described on Page 16 of the results, with methods detailed on Page 31.

We have also added a sentence to the Material and Methods section on Page 31 explaining that DiD labelling was not possible for this experiment, hence our reasoning for taking this alternative approach to detect disseminated tumour cells (highlighted in text on Page 31).

Were the PDX cells transfected with luciferase selected or sorted to ensure 100% expression? If yes, please give details. If not, a mixed population could significantly alter bioluminescent signal readouts as different subpopulations grow out.

The PDX cells transfected with luciferase were not selected to ensure 100% expression. Given the nature of working with PDX cells, we felt that sorting would cause damage to the cells that would be likely to affect their behaviour *in vivo*. Therefore we elected to inject the mixed cell population. In all of our *in vivo* experiments, bone metastases were determined by H&E analyses of bone sections at experiment termination (not by the luminescent signal).

We have explained that we injected an unselected cell population in the Materials and Methods section on Page 31 (highlighted in manuscript text) and have made it clearer that we determined bone metastases by examination of H&E sections in the Materials and Methods on Page 31 (highlighted in manuscript text).

Minor comments:

In response to a reviewer 3 comment about the dormant tumor cells and why only 1/3rd of patients relapse in the bone, an important fact for the authors to keep in mind is that there is a flip side to their argument that IL1b is upregulated. That is that an inhibitory factor may be lost that allows escape from dormancy. For example, there has been some very elegant work showing this is the case for Thrombospondin 1 in the perivascular niche.

We agree with the reviewer, and we have added this to the discussion on Page 21 (highlighted in text). Thank you to the reviewer for making this point.

It could be useful to add the T47D vanctictumab data sent to reviewers to supplementary. If the authors choose to publish the reviews, this will be available anyway.

The T47D mammosphere data has been added as requested, and is now shown in Supplementary Figure 2C. This is described in the text on Page 8 (highlighted in manuscript text).

It is very difficult to see the 1.5x fold change in the figure 4D. The scale should be changed to see this better as there appears to be a lot of heterogeneity so it is not possible to see if it complements the MCF-7 data. In addition to this, have the authors profiled any of the other lines they use in the manuscript?

Figure 4D has now been changed as requested – the axis has been split to show 0-2 fold change as 50% of the axis and make it easier to see the lower fold changes in the data.

We have also profiled the MDA-MB-231_BH cell line response to recombinant IL1 β and have added these data to the MCF-7 results in Figure 4C. Adding this cell line shows that the cell line data mirrors the patient sample data, as in this ER-negative cell line we see less induction of Wnt ligands in response to IL1 β than the ER+ cell line (Wnt 3A and 7A are the only Wnt ligands significantly upregulated in response to IL1 β , and with a reduced fold change compared to MCF-7) which is also what we observed in the patient samples. We would like to thank the reviewer for this suggested experiment, which is a good addition to the manuscript. The manuscript has been modified to include this data and this is shown on Page 13 (highlighted in manuscript text).

Does inhibiting both CREB and NFkB give increased benefit of either alone?

Mammosphere data suggests that there is increased benefit of combined inhibition vs. either inhibitor alone. This is shown in the graph below for both the MCF-7 and MDA-MB-231_BH cell lines (MCF7; CM+sulfasalazine vs CM+Sulfasalazine+KG501 $p < 0.0001$. CM+KG501 vs

CM+Sulfasalazine+KG501 p=0.0020. MDA-MB-231_BH; CM+sulfasalazine vs CM+Sulfasalazine+KG501 p<0.0001, CM+KG501 vs CM+Sulfasalazine+KG501 p<0.0001).

We have added this data to Supplementary Figure 4 and this is discussed in the manuscript text on Page 14 (highlighted in manuscript text), however our onward work will continue to address this question.

The figure 4G western blot appears as though the left-most lane (NFkB p65 Cont) has had a serious issue and not run/transferred/developed correctly. Please provide a better quality blot.

This has been changed for a different blot image (see Figure 4G).

Consider moving the data from 2E to supplementary as it does not really support the story

This has now been moved to Supplementary Figure 2H.

P9L188 – the data presented in the manuscript does not mechanistically support the conclusion that “high-Wnt signaling tumors are those which are more able to colonize the bone environment”. Without the above suggested DKK1 KO *in vivo* study this conclusion needs to be moderated back.

As discussed in a comment above, we have removed this conclusion. We have now grouped all our DKK1 data together, and have concluded “These data add to our previous data indicating that Wnt signalling promotes the ability of CSCs to form colonies in the bone environment *in vitro*, and specifically suggest a role for DKK1 in promoting this, which warrants further investigation in *in vivo* studies.” on Page 10 (highlighted in manuscript text).

P10L206 and L203 – PORNi should probably read PORCNI

Apologies for this typographical error – this has now been changed and is highlighted in the text on Page 11.

P12L257 – “Ascitic” fluids

This has now been changed and is highlighted in the text on Page 13.

In the discussion, the authors could link their discussion of the 30% bone metastases back to their data on DKK1 status. i.e. DKK1 at the primary tumor is likely low in the CSC-like population which then facilitates the metastatic colonization in the bone at a later point.

Thanks to the reviewer for this comment, we have now added this to the discussion on Page 20 (highlighted in manuscript text).

In the methods the authors refer to the MDA213BH and MDS231IV – Are these the same? Be consistent with your naming.

When this cell line was derived it was originally termed “MDA-MB-231_IV”, but we have referred to these cells as MDA-MB-231_BH throughout the manuscript to make it clear to the reader that these are bone homing. We have added this explanation of the change of cell line name to the Materials and Methods section on Page 22 to better explain this. We additionally found two mislabellings of MDA-MB-231_IV in the manuscript text and these have now been changed (highlighted in text). Apologies for this error.

Figure 2D – the contrast of the IVIS photographs is a little difficult to see the mice. The luminescent signal appears OK.

In the version of the Figures we have, the contrast on the IVIS photographs and the mice images are clear. We suspect any issues may therefore be a result of formatting changes during the manuscript submission process. We will check all images with the editorial team for quality and clarify prior to publication of this manuscript to ensure that this is not a problem. Thanks to the reviewer for highlighting this.

Figure 4D – no axis on graph

Apologies for the omission – this has now been added and can be seen in Figure 4D.

Reviewer #4 (Remarks to the Author):

In current manuscript, Eyre and colleagues showed bone marrow culture derived IL1beta induce Wnt autocrine signaling in breast cancer cells thereby increase breast cancer bone colonization. The findings are potentially interesting, however following concerns needs to be addressed:

We would like to thank the reviewer for their interest in this manuscript, and for the support with publication. The reviewer has raised some interesting points which were not discussed in the first round of reviews, and we have responded to these individually below.

It is impressive that the effect of bone marrow condition medium on mammosphere formation is preserved across many early breast cancer patient samples. However, it is not clear why use early breast cancer samples and how these samples relate to bone metastasis? It is also clear how 4 out of 17 samples were chosen for mammosphere self-renewal assay.

We used early breast cancer patient samples throughout the manuscript as an alternative to using breast cancer cell lines. It has been well documented that commercially available cell lines do not well represent primary human cancers (for example in Gillet et al, JNCI. 2013; 105(7): 452–458), therefore we felt using tumour samples from patients at the point of surgery was a more clinically relevant model for examining the response to human tumours to bone marrow secreted factors (and subsequent inhibitors). We have added that these samples were taken at the point of the surgery to the manuscript text on Page 6, to better explain the nature of these samples to readers. We considered our samples particularly clinically relevant as of the 24 samples used in mammosphere experiments during this project, 19 were taken from patients who were treatment naïve at the point of surgery. We have also added this point to the materials and methods section on Page 23 (highlighted in text) to highlight this to readers. We plan to follow up the patients where tumour cells were used in this study and assess which of the patients develop bone metastases, however as metastases in breast cancer often follows a long dormancy phase (particularly for ER+ cancers), follow up for these patients will not be available for several years.

With regards to the 4 patient samples selected for the self-renewal assay, these were the only samples of the 17 used which yielded enough primary mammospheres to replat as secondaries. Typically, primary tumours from breast cancer patients yield a small number of cells following digestion (in a study within our lab of 150 tumour samples digested, we found a mean number of cells isolated of approximately 130,000), and mammospheres form from approximately 0.1-1% of cells plated, therefore not all samples produce enough cells for secondary plating. We have added this information regarding patients to the Materials and Methods section on Page 25. Thanks to the reviewer for pointing this out, as it is an important technical point and was not previously included.

Both mammosphere formation assay and Aldefluor assay supposed to measure breast cancer stem cell property. It is not clear why bone marrow CM has opposite result on MCF7 in these assays.

We completely agree with the reviewer that it is not clear why bone marrow CM has the opposite effect on mammosphere formation vs. ALDH activity. We have also assessed the effect of CM on a further CSC population, CD44+CD24- cells, in the MCF-7 cell line, and did not see an induction of this population with bone marrow CM either (data below). In the example presented here, the percentage of CD44+CD24- cells was 25.9% when cells were treated with control media, vs. 23.9% when cells were treated with bone marrow conditioned media (CM). Results of 3 independent experiments were normalised to control, and no significant difference was seen between control and CM treatment.

A possible explanation for this is that there are multiple different CSC populations which respond differently to bone marrow derived factors. Work from Max Wicha's research group (Liu et al, Stem Cell Reports, 2013;2(1):78-91) has previously shown that there are at least 2 populations of breast CSCs which do not overlap, and have different functional properties. Our results support this, and we show that bone marrow CM induces the specific functional CSC property of colony formation, rather than expression of a particular marker. We have added this to the manuscript on Page 7 (highlighted in text). Following this comment, we have replaced all mentioned of "CSC activity" in the text with "CSC colony forming activity" to better reflect the data, and we have added an additional sentence to the discussion detailing this difference between mammosphere formation and ALDH activity on Page 17. Thank you to the reviewer for this comment, as this has improved clarity throughout the manuscript.

Seems the effect of Vantictumab is limited to *in vitro* and *ex vivo* experiments but not *in vivo* experiments, which does not support the main hypothesis. Furthermore, it is also intriguing that Wnt inhibition increased trabecular bone volume but reduced cortical and spongy bone. It can be helpful to clarify what it means and if it is tumour dependent.

We would like to apologise to the reviewer that the results of our *in vivo* experiments have not been presented clearly enough. Although Vantictumab did not prevent tumour growth following intra-femoral injection (Figure 2), it did prevent bone colonisation *in vivo* following tail-vein injection of MDA-MB-231_BH cells (Figure 5, metastases from 9/16 hind limbs of control mice to 1/14 hind limbs in Vantictumab treated mice (56% vs 7%, p=0.0067)).

To make our *in vivo* results clearer we have moved the intra-femoral tumour formation graph (showing no difference in tumour growth between control and Vantictumab treated mice) from Figure 2 to Supplementary Figure 2H. This has allowed us to place the main focus of Figure 2 on the *ex vivo* effects of Vantictumab, rather than the lack of effect on tumour formation. Hopefully this has made the results of our *in vivo* experiments clearer. We have

also added a discussion regarding the lack of effect of Vantictumab on tumour formation following intra-femoral injection *in vivo* to Page 19 (highlighted in manuscript text).

With regards to the effect of Vantictumab on bone, we used micro-CT to assess the effect of inhibiting Wnt (and IL1 β) signalling on bone biology. We did this as metastatic breast cancer patients often experience bone loss as part of their disease, therefore any treatment which is bone building (in the presence of tumour cells) would be seen as an attractive option for these patients. We found that IL1 β inhibition increased trabecular bone volume, which has highlighted this to us as a potentially particularly useful treatment in metastatic breast cancer. Vantictumab, on the other hand, has been previously shown to have negative effects on bone biology in clinical trials (although this can be reversed with concurrent treatment with Zoledronic acid), and we wanted to see if this was also true in mice with bone metastases. Our micro-CT data supported this, as negative bone effects can be seen in the images in Figure 5D. However, in the quantification and graphical representation of BV/TV (Supp Figure 5E) it looked as if trabecular bone volume increased, but this was due to a loss of bone volume overall rather than a bone-building effect. To make this clearer, we have also now analysed cortical bone volume in untreated vs. Vantictumab treated mice, and have shown that cortical bone volume is significantly reduced with the anti-wnt treatment ($p < 0.0001$, $n = 7$) (graph below). We have added this to the manuscript in Supp Figure 5F, and we have reworded the text around this on Page 15 of the manuscript to improve clarity (highlighted in text). We have also added a better description of why we performed the micro-CT analysis, and this is on Page 15 (highlighted in text).

Seem the *in vivo* effects of IL1beta inhibition and Wnt inhibition are different. IL1beta inhibition seems to affect bone homing while Wnt inhibition does not. This seems contradict with the authors' model that these two pathways work in a linear fashion. Furthermore, the fact that Wnt inhibition leads to dormancy *in vivo* needs to be proved directly as the effect can be also explained by reduced proliferation.

This comment links to the one above, and hopefully with our improved clarification of the Vantictumab results in the text, the reviewer will be happier with our conclusion than Wnt inhibition does prevent bone colonisation *in vivo* (see Figure 5). We agree with the statement that Wnt inhibition does not affect bone homing, as we have shown in our multi-photon

analysis (Figure 5E) that Vancitumab treatment does not prevent tumour cell arrival in the bone (as a similar number of DiD labelled cells were present in the bone on Vantictumab and control mice at 28 days despite less overt metastases forming in Vantictumab treated mice). In a previous publication, we showed similar results for IL1 β inhibition with Anakinra (Holen et al, Oncotarget 2016; 7 (46) 75571-75584), and we have completed further experiments staining bones from control and Anakinra treated mice without overt metastases from our PDX experiment (Figure 5F) with cytokeratin 19 to assess the presence of tumour cells in these bones (example of this immunohistochemical staining shown below, disseminated tumour cells detected using phycoerythrin conjugated cytokeratin 19 (sc376126) (red), bone shown in blue).

Red cancer cells could be detected in the bones of Anakinra treated mice, showing that, as with the cell line experiments, tumour cells reached the bones, but did not grow into overt metastases. This provides evidence to suggest that IL1 β and Wnt inhibition may have similar effects, although this is something we are investigating in our onward work. We have added these data to the manuscript in Supplementary Figure 5H, and this is described on Page 16 of the results, with methods detailed on Page 31.

We agree with the reviewer that although we have shown that Wnt inhibition does not prevent tumour cell homing, further experimental work would be required to conclude that Wnt inhibition leads to dormancy *in vivo*. Therefore we have removed the use of the term “dormant” on Page 15, and have instead rephrased this as “28 days following injection, similar numbers of tumour cells were present in the proximal tibiae of control or Vantictumab treated mice (Figure 5E), demonstrating that although inhibiting Wnt signalling prevents colonisation of cells to form bone metastasis, it does not prevent tumour cells homing to bone”, which better reflects the experimental data we have. Thank you to the reviewer for this constructive comment to improve the manuscript.

It is critical to show if the proposed model is relevant to the disease in patients. The authors showed that DKK level is associated with bone metastasis free survival. However, it can be important to show if there is any connection between IL1beta and DKK1/Wnt pathway in patient dataset and if IL1beta level is also associated with bone metastasis free survival in order to support the proposed model?

The connection between IL1 β and Wnt in patients is an interesting question. In this manuscript we show a connection between DKK1 and bone metastases in patient datasets, and in a previous publication we showed that IL1 β expression in primary tumours predicts bone metastasis (Tulotta et al, Clinical Cancer Research 2019; 25 (9) 2769-2782). Between these two publications we have therefore shown that both parts of this identified pathway are relevant to patient breast cancer. We have included a reference to this IL1 β patient data in the manuscript discussion on Page 18 (highlighted in manuscript text) to make this existing clinical data clearer.

In response to the reviewer's specific question, we have performed further analysis looking for links between DKK1 and IL1 β in breast tumours in patient datasets (METABRIC, TCGA, NKI295). No significant correlations were observed (data below), however given the hypothesis of this study that the IL1 β driving metastatic colonisation is microenvironmental, and this activates Wnt signalling in a sub-population of breast CSCs, we do not think this lack of correlation is unexpected. The role of tumour derived vs. microenvironmental IL1 β at metastatic sites forms part of our next work on this topic, as increasing literature on IL1 β suggests the role for this cytokine in promoting tumour formation and metastasis is complex.

REVIEWERS' COMMENTS:

Reviewer #2 (Remarks to the Author):

In this revised version, the authors have addressed my comments and the manuscript is improved. The study is interesting and timely and of wide interest. Therefore, I now recommend publication.

Reviewer #3 (Remarks to the Author):

The authors have made several additions and changes to this revised manuscript in response to all of the reviewer comments, including a valuable discussion on the limitations and strengths of their models/approaches and refined their conclusions so that they support the wealth of data they now present on homing/colony forming ability/dormancy/mechanism etc. I would like to thank the authors for the time and care spent in revising the manuscript, which is exciting and well balanced in their interpretations and a pleasure to read.

The addition of the data showing induction of IL1b in BM by tumor CM has added strength to the manuscript, as has the new link to the authors recent Tulotta Clin Can Res paper 2019, further highlighting a novel interplay between IL1b, DKK and overt bone metastasis

The rearrangement of the DKK1 data and addition of the Q-RT supports their conclusions and makes the manuscript easier to understand.

The clarification on the uCT bone, PORCNi, IL1R+ enrichment and PDX-Luc bone metastases methodology is greatly appreciated and will also help ensure readers understand the experimental approaches and data presented.

The additional PDX disseminated tumor cell data (Sup 5H) is very helpful, as in the clarification on why DiD imaging was not possible.

I accept the authors point (and preliminary data) with regard to IL1b immune modulation in the 4T1 and E0771 models. I am in agreement with the authors that a comprehensive dissection of the immune-dependent role would be beyond the scope of the current manuscript, and feel that the new data that they have chosen to include is enough to support their interpretations since they discuss this in the context of recent publications in the field.

Given the significant improvements that have been made and the timely nature of this work, I have no hesitation in recommending this work for immediate publication.

Very minor points:

In Figure 1G, it is difficult to see the wound at 18 hours and appears as though it has completely closed in CONT, but not in CM, something which is not reflected in the graph. Perhaps a white overlay tracing the wound edge would be helpful here?

In Figure 2H it appears that the error bars are missing from the MDA-MB-231 data.

In Figure 3A, could the color map be adjusted so that the black is less dominant and the color scheme is deuteranopia-friendly. I would suggest yellow-blue with a 20-25% black center-point as this would help see the differences more clearly, especially when printed.

In Figure 3C, the error bars appear to be missing, as do the statistics.

In Figure 5A, C and F, the error bars appear to be missing.

It would be helpful to add the median survivals and HRs to Supp Fig 2I

Reviewer #4 (Remarks to the Author):

Seems current manuscript has significantly improved and addressed my concerns. I don't have additional comments.

REVIEWERS' COMMENTS:

Reviewer #2 (Remarks to the Author):

In this revised version, the authors have addressed my comments and the manuscript is improved. The study is interesting and timely and of wide interest. Therefore, I now recommend publication.

We thank the reviewer for the constructive comments they made to improve the manuscript, and for their support in publishing these findings.

Reviewer #3 (Remarks to the Author):

The authors have made several additions and changes to this revised manuscript in response to all of the reviewer comments, including a valuable discussion on the limitations and strengths of their models/approaches and refined their conclusions so that they support the wealth of data they now present on homing/colony forming ability/dormancy/mechanism etc. I would like to thank the authors for the time and care spent in revising the manuscript, which is exciting and well balanced in their interpretations and a pleasure to read.

The addition of the data showing induction of IL1b in BM by tumor CM has added strength to the manuscript, as has the new link to the authors recent Tulotta Clin Can Res paper 2019, further highlighting a novel interplay between IL1b, DKK and overt bone metastasis

The rearrangement of the DKK1 data and addition of the Q-RT supports their conclusions and makes the manuscript easier to understand.

The clarification on the uCT bone, PORCNi, IL1R+ enrichment and PDX-Luc bone metastases methodology is greatly appreciated and will also help ensure readers understand the experimental approaches and data presented.

The additional PDX disseminated tumor cell data (Sup 5H) is very helpful, as in the clarification on why DiD imaging was not possible.

I accept the authors point (and preliminary data) with regard to IL1b immune modulation in the 4T1 and E0771 models. I am in agreement with the authors that a comprehensive dissection of the immune-dependent role would be beyond the scope of the current manuscript, and feel that the new data that they have chosen to include is enough to support their interpretations since they discuss this in the context of recent publications in the field.

Given the significant improvements that have been made and the timely nature of this work, I have no hesitation in recommending this work for immediate publication.

Very minor points:

In Figure 1G, it is difficult to see the wound at 18 hours and appears as though it has completely closed in CONT, but not in CM, something which is not reflected in the graph.

Perhaps a white overlay tracing the wound edge would be helpful here?
An overlay tracing the wound edge has been added to Figure 1G.

In Figure 2H it appears that the error bars are missing from the MDA-MB-231 data.
This PCR data was normalised to MDA-MB-231 for each repeat, and the MDA-MB-231_BH are fold changes from the parental MDA-MB-231 cell line (n=6 repeats). The individual data points are now shown on the figure to better reflect this.

In Figure 3A, could the color map be adjusted so that the black is less dominant and the color scheme is deuteranopia-friendly. I would suggest yellow-blue with a 20-25% black center-point as this would help see the differences more clearly, especially when printed.
This figure has now been changed to a yellow-blue colour scheme. Thanks to the reviewer for this point.

In Figure 3C, the error bars appear to be missing, as do the statistics.
There are no error bars on this graph as this is a quantification of the single blot (n=1) shown. This has been added to the figure legend to make this clearer.

In Figure 5A, C and F, the error bars appear to be missing.
In all of these figures, the graphs represent a quantification of the percentage of mice in each group with tumours. These are single percentages summarising the result of each experiment (experiments containing n=16, n=16 and n=24 mice respectively), therefore no error bars are present. This is explained in the text relating to Figure 5, and Chi squared tests (tumour vs. no tumour in treated vs. untreated mice) have been performed on these data. This is detailed in the Figure legend.

It would be helpful to add the median survivals and HRs to Supp Fig 2I
Hazard ratios have been added to Supp Fig 2I and Figure 2F.

Reviewer #4 (Remarks to the Author):

Seems current manuscript has significantly improved and addressed my concerns. I don't have additional comments.

We thank the reviewer for their support in publishing these findings.